



# Identification of source regions of the Asian Tropopause Aerosol Layer on the Indian subcontinent in August 2016

Jan Clemens[1,2,3], Bärbel Vogel[1,3], Lars Hoffmann[2,3], Sabine Griessbach[2,3], Nicole Thomas[1,3], Suvarna Fadnavis[4], Rolf Müller[1,3], Thomas Peter[5], and Felix Ploeger[1,3,6]

[1]Institut für Energie- und Klimaforschung (IEK-7), Forschungszentrum Jülich, Jülich, Germany
[2]Jülich Supercomputing Center (JSC), Forschungszentrum Jülich, Jülich, Germany
[3]Center for Advanced Simulation and Analytics (CASA), Forschungszentrum Jülich, Jülich, Germany
[4]Indian Institute of Tropical Meteorology (IITM), Pune, India
[5]Institute for Atmospheric and Climate Science (IAC), Swiss Federal Institute of Technology (ETH), Zürich, Switzerland
[6]Institute for Atmospheric and Environmental Research, University of Wuppertal, Wuppertal, Germany.

**Correspondence:** j.clemens@fz-juelich.de

**Abstract.** The Asian tropopause aerosol layer (ATAL) is a distinct feature during the Asian summer monsoon season with an impact on the regional radiative balance of the Earth's atmosphere. However, the source regions and the detailed transport pathways of ATAL particles are still uncertain. In this study, we investigate transport pathways from different regions at the model boundary (MB) to the ATAL using the two Lagrangian transport models CLaMS (Chemical Lagrangian Model of the Stratosphere) and MPTRAC (Massive-Parallel Trajectory Calculations), two reanalyses (ERA5 and ERA-Interim), and balloon-borne measurements of the ATAL performed by the Compact Optical Backscatter Aerosol Detector (COBALD) above Nainital (India) in August 2016. Trajectories are initialized at the location of the ATAL, as measured by COBALD in the Himalayas, and calculated 90 days backward in time to investigate the relation between the measured, daily averaged, aerosol backscatter ratio and different source regions at the MB. Nine source regions at the MB are distinguished, marking continental and maritime sources in the region of the Asian monsoon. Different simulation scenarios are performed, to find systematic differences as well as robust patterns, when the reanalysis data, the trajectory model, the vertical coordinate (kinematic and diabatic approach) or the convective parameterisation are varied.

While there are many robust features, the simulation scenarios also show some considerable differences between the air mass contributions of different source regions at the MB in the region of the Asian monsoon. The contribution to all air parcels from the MB varied between 5% and 40% for the Indo-Gangetic plain, the contribution from the Tibetan Plateau varied between 30% and 40% and contributions from oceans varied between 14% and 43% for different scenarios. However, the robust finding among all scenarios is that the largest continental air mass contributions originate from the Tibetan plateau and the India subcontinent (mostly the Indo-Gangetic plain), and largest maritime air mass contributions in Asia come from the Western Pacific (e. g. related to tropical cyclones such as typhoons). Additionally, all simulation scenarios indicate that transport of maritime air from the Tropical Western Pacific to the region of the ATAL lowers the backscatter ratio (BSR) of the ATAL, while most scenarios indicate that transport of polluted air from the Indo-Gangetic plain increases the BSR. Therefore, while the results corroborate key findings from previous ERA-Interim based studies, they highlight the variability of the contributions





of different MB regions to the ATAL depending on the meteorological input data, vertical velocities and in particular on the treatment of convection within the model calculations.

# 1 Introduction

The Asian tropopause aerosol layer (ATAL) is a layer of particles over Asia in the upper troposphere and lower stratosphere (UTLS) during the Asian monsoon. The source regions and the chemical composition of the ATAL particles are subject of current debate. The ATAL occurs between May and September, with a peak in summer in coexistence with the Asian summer monsoon anticyclone (ASMA) (Brunamonti et al., 2018). The ATAL extends from around 15° N to 35° N and 0° E to 150° E at

heights between 14 km and 18 km, even though the extent and density of the ATAL shows a distinct variability on time scales of days, months and years (e.g. Vernier et al., 2011; Hanumanthu et al., 2020). First evidence of the ATAL from balloon-borne observations was found in 1999 (Kim et al., 2003; Tobo et al., 2007). The large extent of the ATAL was later demonstrated by satellite observations (Vernier et al., 2011; Thomason and Vernier, 2013).

Even though the first satellite observations indicated that the ATAL aerosol particles are liquid, i.e., they are identified as

spherical objects, or very small solid particles (e.g. dust) (Vernier et al., 2011), more direct conclusions about the chemical composition and possible source regions of the ATAL were originally not possible. Unique aircraft measurements over the Indian subcontinent in summer 2017 gave deeper insights into the chemical composition of ATAL particles indicating that ammonium, nitrate and organics are important contributors to the chemical composition of ATAL particles (e.g. Höpfner et al., 2019; Appel et al., 2022). Appel et al. (2022) highlight that a significant particle fraction within the ATAL results from the

conversion of gas-phase precursors, rather than from the uplift of primary particles from below. Furthermore, Weigel et al. (2021) found evidence of new particle formation at ATAL altitudes emphasizing the presence of secondary aerosol.

To identify the transport pathways as well as the source regions on the Earth's surface for air masses that contribute to the ATAL or the Asian monsoon anticyclone a variety of studies were performed (e.g. Bergman et al., 2012; Vogel et al., 2015; Höpfner et al., 2019; Hanumanthu et al., 2020; Fairlie et al., 2020). The transport in the Asian summer monsoon region is

determined by deep convection, which injects air masses from the surface rapidly into the UTLS (up to 360K) within a few hours. In the presence of the ASMA, air masses are transported upward by slow diabatic heating (with a vertical velocity of about 1-1.5 K per day in ERA-Interim) superimposed on the anticyclonic flow resulting in an upward spiraling movement of individual air parcels (Vogel et al., 2019). Deep convection contributing to transport in the ASMA has been reported from a wide range of regions, such as South India, the Bay of Bengal or China (e.g. Vernier et al., 2018; Bucci et al., 2020; Zhang et al.,

2019a, b, 2020), following a wide variety of convective activity over Asia (Fadnavis et al., 2013). In addition, the transport from the Western Pacific's boundary layer into the UTLS within typhoons and subsequent transport into the ASMA was shown (Vogel et al., 2014; Li et al., 2017, 2020; Hanumanthu et al., 2020). However, most of the air is injected by continental convection into the ASMA at the southern edge of the Himalayas (Indo-Gangetic plain, Foothills, Tibetan plateau) (Bergman et al., 2012; Fadnavis et al., 2017; Höpfner et al., 2019; Bucci et al., 2020). Additionally, the relation between source regions

at the model boundary (MB) and the aerosol backscatter intensity of the ATAL has been studied. Evidence from balloon-



borne and aircraft measurements indicates that if air from the ocean is injected into the ATAL, the BSR is reduced, whereas it increases for larger continental contribution (Hanumanthu et al., 2020).

Lagrangian trajectory calculations in combination with observations have been used to investigate the relation between source regions at the Earth's surface and the chemical composition of air masses within the Asian monsoon anticyclone as well as ATAL properties (e.g. Li et al., 2017; Vernier et al., 2018; Höpfner et al., 2019; Legras and Bucci, 2020; Johansson et al., 2020; Hanumanthu et al., 2020; Zhang et al., 2019b). These calculations rely on reanalysis data and their ability to correctly resolve convection and the diabatic vertical ascent in the ASMA. The ERA-Interim reanalysis (Dee et al., 2011) was frequently used in previous Lagrangian transport studies of the ATAL. The ERA5 reanalysis (Hersbach et al., 2020) is the latest reanalysis provided by the European Centre for Medium-Range Weather Forecasts (ECMWF) and has replaced the ERA-Interim reanalysis since 2019. ERA5 has a much higher temporal and spatial resolution than ERA-Interim which impacts atmospheric transport simulations (Hoffmann et al., 2019) and improves substantially the resolution of convection and tropical cyclones (e.g. typhoons) (e.g. Li et al., 2020; Taszarek et al., 2020; Malakar et al., 2020).

In studies of the source regions that are contributing to the composition of the Asian monsoon anticyclone, better agreement between diabatic and kinematic calculations and between models and observations was found when ERA5 was used instead of ERA-Interim (e.g. Bucci et al., 2020; Legras and Bucci, 2020). The attribution of the sources, however, still depends on the reanalysis data (Bucci et al., 2020). Vertical transport from the MB to the UTLS is faster with ERA5 than with ERA-Interim (e.g. Li et al., 2020). Altogether, these studies indicate, that ERA5 improves the simulations in comparison to the ERA-Interim reanalysis.

Lagrangian transport calculations are expected to be well suited for the detection of ATAL surface source regions. However only a few investigations have been done with regard to the robustness of this approach against variation of the reanalysis data, transport models and vertical velocities. In this study, we build upon the work of Hanumanthu et al. (2020) to reevaluate the source regions of the ATAL based on backward trajectories and balloon-borne measurements in Nainital, India in 2016. Hanumanthu et al. (2020) investigated ATAL source regions of COBALD balloon-borne measurements using CLaMS (Chemical Lagrangian Model of the Stratosphere) diabatic backward trajectory calculations driven by ERA-Interim. Here, we extend this approach with different simulation scenarios based on two reanalyses (ERA5 and ERA-Interim), two Lagrangian transport models (Chemical Lagrangian Model of the Stratosphere and Massive-Parallel Trajectory Calculations), two types of vertical velocities (diabatic and kinematic trajectories), and with changes of integration time-steps and parameterisation parameters (e.g. for convection). The goal of these sensitivity tests is to identify differences and robust transport features that emerge from different simulation setups for the vertical transport, including explicit and parameterized convection.

We provide more detailed descriptions of the data, methods, definitions and simulation scenarios in Sect. 2. In Sect. 3, we illustrate typical transport pathways to the locations of the measurements over Nainital. Next, we discuss the vertical and horizontal distribution of the air parcels over different heights and regions. Then, we determine the temporal characteristics of the transport for specific regions. Finally, the correlation between the daily ATAL backscatter intensity and source regions is investigated. The summary and conclusions are presented in Sect. 4.





## 2    Data and methods

This study follows closely the approach of Hanumanthu et al. (2020). Balloon-borne measurements with a backscatter sonde allowed to identify the location of the ATAL along the balloon ascents. Using the Lagrangian transport models, air parcels can be initialized at the detected ATAL locations and transported backward to the model boundary (MB) to find possible source regions of the ATAL. Additionally, the measured and daily averaged aerosol backscatter at the location of the balloons can be related to surface regions and their properties. In this section, the methods of Hanumanthu et al. (2020) are described in more

detail and novel elements that are applied in this study are elaborated.

### 2.1    COBALD aerosol measurements

In August 2016, 15 balloons were launched in Nainital, Uttarakhand, India (29.35°N, 79.46°E, 1820 m above sea level) (Brunamonti et al., 2018). The balloons carried the Compact Optical Backscatter Aerosol Detector (COBALD), which is a lightweight

backscatter sonde (Brabec et al., 2012). It measures the backscatter at 940 nm (infrared) and 455 nm (blue visible) in proximity to the balloon. For the detection of the ATAL the short wavelenghth channel (455 nm) is used (details see Hanumanthu et al., 2020). Furthermore, the balloon carried a RS41-SGP radiosonde that logged local temperature and pressure. The backscatter signal can be expressed as the backscatter ratio (BSR). The BSR is the ratio between the total backscatter due to aerosols and air molecules and the backscatter due to air molecules alone. Based on calculations of Bucholtz (1995), the BSR has been inferred

from the temperature and pressure of the radiosonde and the backscatter of COBALD. Furthermore, a color index (CI), i. e., the ratio between the 940 nm and 455 nm aerosol backscatter has been calculated, because it allows us to discriminate between large aerosol particles and smaller ones and accordingly between layers of cirrus clouds and ice-free layers of the ATAL. This analysis provided vertical profiles of the aerosol layer for 15 days in August 2016. Table 1 gives a short overview over the measurements. While the BSR is determined for every measurement point during the ascent, here we use the daily, vertical

average of the BSR for each balloon flight. Hence, we analyse the day to day changes of the measured BSR profiles and do not analyse the BSR for every measurement time-step individually. For further details see Hanumanthu et al. (2020).

### 2.2    Lagrangian transport models

Lagrangian backward trajectories are started at the COBALD observations of the ATAL to identify the location and time when air parcels contributing to ATAL were released at the model boundary. For the backward trajectory calculations, two different

Lagrangian models were used: The Chemical Lagrangian Transport Model of the Stratosphere (CLaMS) and the Massive-Parallel Trajectory Calculations (MPTRAC) model.

CLaMS is a full chemical Lagrangian transport model that includes modules for irreversible mixing, chemistry and advection (McKenna et al., 2002b, a). Here, we focus on the advection module alone, which applies a 4th-order Runge-Kutta scheme for the trajectory calculations with a default integration time-step of 1800 s. The CLaMS model can be used in hybrid vertical

coordinates (Mahowald et al., 2002; Ploeger et al., 2010; Pommrich et al., 2014; Ploeger et al., 2021). The hybrid coordinate $\zeta$ is near the surface an orography-following sigma coordinate and transforms continuously into potential temperature at higher





altitudes above around 300 hPa and 380 K, respectively. The vertical velocity can be calculated by the reanalysis diabatic heating rates (diabatic vertical velocity) or by the reanalysis vertical velocities from mass balance (kinematic vertical velocity). CLaMS can be used with both vertical velocity approaches (e.g. Ploeger et al., 2010; Li et al., 2020), however, in most CLaMS studies the diabatic approach is used.

MPTRAC (Hoffmann et al., 2016, 2022) is a Lagrangian transport model for the free troposphere and the stratosphere. It includes modules for advection, diffusion, and convection, which are applied in this study. The advection module uses the mid-point scheme for integration with a default time-step of 180 s (Rößler et al., 2018). MPTRAC has been further developed for this study to use either pressure or zeta coordinates (kinematic or diabatic vertical velocities) to calculate the trajectories following the approach in CLaMS. In contrast to CLaMS, mixing is computed with two modules that parameterize diffusion and subgrid-scale winds, using given diffusivity coefficients and parameterized subgrid-scale wind fluctuation (Stohl et al., 2005).

Additionally, MPTRAC implements the extreme convection parameterisation (ECP) to represent the effects of unresolved convection in the reanalysis data, based on the work of Gerbig et al. (2003). The ECP vertically mixes the air parcels within a convective column by a randomized density-weighted distribution between the surface and the equilibrium level (EL). The stability is assessed with the convective available potential energy (CAPE) and the convective inhibition (CIN). CAPE is the integrated amount of work that the upward buoyancy force would perform on a given mass of air if it rose vertically through the atmosphere. The CIN is the energy that air parcels need to overcome when a stable layer below the level of free convection exists.

## 2.3 Reanalysis data

We used full-resolution ERA5, downsampled ERA5 and ERA-Interim reanalysis data to drive the backward trajectory calculations with CLaMS and MPTRAC. Both reanalyses have been developed by the ECMWF. ERA-Interim is the precursor of ERA5. The ERA-Interim reanalysis offers six-hourly meteorological data at around 80 km horizontal resolution on 60 levels. It reaches from the surface up to 0.1 hPa. The ERA-Interim reanalysis is available for the years from 1979 to 2019. The assimilation system for ERA-Interim uses a four-dimensional variational analysis (4D-Var) with a 12 h time window and the ECMWF's Integrated Forecast System (IFS) cycle 31r2 as released in 2006.

The ERA5 reanalysis offers hourly meteorological data on a 30 km horizontal grid ($0.3° \times 0.3°$) over 137 levels from the surface up to 80 km. The ERA5 reanalysis was processed with an improved model version compared to ERA-Interim (IFS cycle 41r2 with 4D-Var assimilation), including novel parameterisations of atmospheric waves and convection. The ERA5 reanalysis covers the time period between 1950 and the present. Increase of spatial and temporal resolution in ERA5 particularly improves the representation of tropical cyclones and convection in the reanalysis in comparison to ERA-Interim and other reanalysis data (e.g. Taszarek et al., 2020; Li et al., 2020). ERA5 was also found to significantly improve Lagrangian transport simulations in the free troposphere and stratosphere (Hoffmann et al., 2019).

The low resolution ERA5 data set (referred to as ERA5lr) was created by down-sampling of the full-resolution data to a $1° \times 1°$ horizontal grid and 6 hour time-steps, applying a truncation to T213 as is specified in the ECMWF MARS data





processing system. The vertical levels of ERA5 were kept unchanged. Low-resolution ERA5 data was used in previous studies to benefit from the improvements of the ERA5 reanalysis but to avoid high computational costs and costs for handling the much larger amount of data compared to ERA-Interim (e.g. Ploeger et al., 2021).

## 2.4 Simulation scenarios

For the Lagrangian backward trajectory calculations with CLaMS and MPTRAC, the air parcels are initialized at positions of the measured ATAL in August 2016 on 15 measurement days. For every measurement time of the COBALD instrument, i.e. every second, one air parcel is initiated. During a flight the balloons horizontal drift is below 10 km in the ATAL. The differences are below 50 km in the ATAL when different balloons, from different days, are compared and hence, they are as well negligible. Table 1 shows an overview over the measurements and the number of air parcels initialized per day. Two

balloon flights are discussed separately; the flight on 12 August, when a large cirrus cloud covered the full UTLS in the sampled region and 15 August, when no ATAL was detected (for details see Hanumanthu et al., 2020). All air parcels are each calculated backward for 90 days to cover the entire Asian monsoon period (JJA).

Different scenarios for the calculations have been applied to study the impact of the reanalysis data (ERA5 vs. ERA-Interim), Lagrangian model differences (CLaMS vs. MPTRAC), vertical velocities (diabatic and kinematic), modules and parametriza-

tions (convection vs. no convection) and the size of the time-step (180 s vs. 1800 s) on the simulated transport. We consider the default configurations of the models for the integration time-step. Additionally, one scenario with MPTRAC is included, for which instead of initialising one air parcel per measurement point, 1000 air parcels are initialized and random perturbations along the trajectories were added. With this ensemble approach, sampling uncertainties were estimated. For the particle diffusion we used the default settings (see Hoffmann et al., 2022). A summary of all scenarios can be found in Table 2.

**Table 1.** Overview on COBALD measurements in August 2016. $\overline{\mathrm{BSR}}$ is the daily, vertically averaged BSR. The number of measurements i.e. the number of initialized air parcels per flight is labeled #AP.

| day | 02 | 03 | 05 | 06 | 08 | 11 | 12 | 15 | 17 | 18 | 19 | 21 | 23 | 26 | 30 |
|-----|-----|-----|-----|-----|-----|-----|-----|-----|-----|-----|-----|-----|-----|-----|-----|
| $\overline{\mathrm{BSR}}$ | 6.7 | 9.2 | 6.7 | 8.3 | 7.1 | 7.0 | - | 2.3 | 7.6 | 6.5 | 7.3 | 5.6 | 5.4 | 8.0 | 5.9 |
| #AP | 670 | 419 | 413 | 385 | 680 | 269 | 463 | 444 | 651 | 705 | 569 | 250 | 331 | 120 | 392 |



**Table 2.** Overview over scenarios of 90 days backward calculations performed for the ATAL measurements above Nainital in August 2016. The abbreviation for each scenario contains at the first position the reanalysis, at the second position the vertical velocity and at the third position the model, where each label is separated by a dash. Optional properties are added the same way at the last position.

| abbreviation | reanalysis | vertical velocity | model | time-step | convection parameterisation | dispersion |
|---|---|---|---|---|---|---|
| EI-kin-C | ERA-Interim | kinematic | CLaMS | 1800s | no | no |
| EI-kin-M | ERA-Interim | kinematic | MPTRAC | 180s | no | no |
| EI-dia-C | ERA-Interim | diabatic | CLaMS | 1800s | no | no |
| EI-dia-M | ERA-Interim | diabatic | MPTRAC | 180s | no | no |
| E5-kin-C | ERA5 | kinematic | CLaMS | 1800s | no | no |
| E5-dia-C | ERA5 | diabatic | CLaMS | 1800s | no | no |
| E5-kin-M | ERA5 | kinematic | MPTRAC | 180s | no | no |
| E5-dia-M | ERA5 | diabatic | MPTRAC | 180s | no | no |
| E5-kin-M-ECP | ERA5 | kinematic | MPTRAC | 180s | yes | no |
| E5-kin-M-1800s | ERA5 | kinematic | MPTRAC | 1800s | no | no |
| E5-kin-M-Diff | ERA5 | kinematic | MPTRAC | 180s | no | yes |
| E5lr-kin-M | ERA5 low res. | kinematic | MPTRAC | 180s | no | no |
| E5lr-dia-M | ERA5 low res. | diabatic | MPTRAC | 180s | no | no |

## 2.5 Classification of air parcel origin

The origin of the air parcels found in the ATAL is classified vertically and horizontally based on the 90 day backward trajectory calculations. Vertically, we follow Hanumanthu et al. (2020) with four classes, the model boundary (MB), the lower troposphere (LT), the upper troposphere (UT) and the lower stratosphere (LS). These vertical layers are defined by values of the vertical coordinate $\zeta$ and the potential temperature $\theta$ as presented in Table 3. The MB is defined as the layer below the 120 K $\zeta$-level, which approximately corresponds to heights between 2 km and 3 km. Accordingly, an air parcel is considered to originate from the MB if it is located at any time below the 120 K $\zeta$-level (Pommrich et al., 2014; Vogel et al., 2019; Hanumanthu et al., 2020).

Hanumanthu et al. (2020) used backward trajectory times between 40 and 80 days. We extended the calculations to 90 days of time for each air parcel to completely cover the entire Asian summer monsoon season in our analysis. We found that a majority of the transport from the MB to the ATAL occurred within 90 days (i.e. 60%-90% of air parcels reach the MB during that time), with only low increase when longer integration time is used (lower than around 5 percentage points per 10 additional days).

When an air parcel is classified as originating from the MB, it is also horizontally classified according to the position where it left the MB for the last time. For this position, we defined several possible source regions. The regions were motivated by different surface characteristics, such as the presence of aerosol or aerosol precursors and by source regions proposed by earlier





studies (e.g. Li et al., 2017; Höpfner et al., 2019; Li et al., 2020; Hanumanthu et al., 2020). For the continents, the following regions are defined: the Asian Highlands, i.e. mostly the Tibetan Plateau, the Indo-Gangetic plain together with the foothills of the Himalayas, a region South of India plus Sri Lanka and finally South East Asia. Other parts of the continents are summarized as the residual continent. For the oceans, three regions were defined: the Tropical Western Pacific that has been affected by typhoon activity, the Arabian Sea, the Bay of Bengal and the residual oceans. Figure 3a illustrates the different regions. For

more detail of the definitions, see Appendix A.

Additionally, for those air parcels that do not originate from the MB, but instead circulate still in the ASMA after 90 days of backward trajectory time, we define the class of the ASMA. For this purpose, we are considering the 3D box from $0^{\circ}$ E to $135^{\circ}$ E, and from $0^{\circ}$ N to $45^{\circ}$ N (magenta box in Fig. 3a) within the UTLS region. Each air parcel within this box is considered to be part of the ASMA.

**Table 3.** Classes for vertical classification of the distributed air parcels. Air parcels have to fulfil all he criteria to be attributed to a specific class. $\zeta$ is the vertical zeta-coordinate, $\theta$ is the potential temperature, $\lambda$ is the longitude and $\phi$ is the latitude of an air parcel.

| class | $\zeta$-criterion | $\theta$-criterion | lon/lat criterion | abbrev. |
|---|---|---|---|---|
| Model boundary | $\zeta \leq 120\,\mathrm{K}$ | | | MB |
| Lower troposphere | $\zeta > 120\,\mathrm{K}$ | $\theta \leq 340\,\mathrm{K}$ | | LT |
| Asian summer monsoon anticyclone | $\zeta > 120\,\mathrm{K}$ | $\theta > 340\,\mathrm{K}$ | $0 \leq \lambda \leq 135°, 0 \leq \phi \leq 45°$ | ASMA |
| Upper troposphere | $\zeta > 120\,\mathrm{K}$ | $340\,\mathrm{K} < \theta \leq 370\mathrm{K}$ | not in ASMA | UT |
| Lower stratosphere | $\zeta > 120\,\mathrm{K}$ | $370\,\mathrm{K} > \theta$ | not in ASMA | LS |




## 3 Results

In the following, we present transport pathways, transport times and possible surface source regions of air mass contributions to the ATAL above Nainital in August 2016. Furthermore, the relation between the observed ATALs backscatter intensity and different model boundary regions is analysed. Our analysis is performed for different simulation scenarios as described in Sect. 2.4.

### 3.1 Transport pathways from source regions to the measured ATAL

The ASMA extends from northeast Africa to the Western Pacific from early June until the end of September, therefore air parcels circulate in the ASMA over a wide range of longitudes and latitudes. Depending on its extension and position, convection can uplift air from different regions of the Earth's surface - i.e. with different chemical composition - into altitudes of the anticyclone. Within the ASMA, the air from different origins will be mixed, for example due to instabilities of the ASMA (e.g. Gottschaldt et al., 2018). Different source regions can contribute to the chemical composition of the ASMA, therefore trace gases and aerosol are in general not homogeneously distributed within the ASMA. The ASMA can show a bimodal structure, where one circulation centre is placed roughly over Iran and the other one is centred roughly over South-East-Asia and where the separation of the two modes varies in time (Yongfu et al., 2002; Nützel et al., 2016; Honomichl and Pan, 2020).

Figure 1 shows four exemplary transport pathways of air parcels from different regions at the model boundary to the measured ATAL over Nainital in August 2016 due to convective transport. For the illustration of transport pathways we use the scenario E5-kin-C, because it is representative with regard to the general patterns also for the other scenarios, while particular differences are analysed in depth later. We also include trajectories of all days of measurement for the illustration.

Figure 1a shows injections into the center of the ASMA in proximity of Nainital, originating from the Tibetan plateau. Throughout the season the air is pumped upward into the ASMA on timescales from hours to a few days (50% of transport from the MB into the UTLS is within three days, see also Tables E2 and E1). In the ASMA, air masses are uplifted by diabatic heating superimposed by the anticyclonic flow until they meet the measurement points over Nainital. Air parcels circle in a rising spiral, within the two modes of the ASMA, until they meet the measurement points over Nainital (e.g. Vogel et al., 2019). A small number of air parcels leaves the ASMA for a while, and is subsequently transported along the subtropical jet, circumnavigating the Earth, until the air parcels are trapped in the ASMA again. Thereafter, the air parcels also arrive at the measurement points.

Figure 1b shows the uplift of air into the ASMA mostly over the Indo-Gangetic plains and at the foothills of the Himalayas. The air masses were transported mainly directly from the Indo-Gangetic plain into the UTLS. The transport in the ASMA and sporadically along the jet-streams is the same as for the Tibetan Plateau. Hence, at the foothills of the Himalayas, transport pathways from two regions with two very different land-cover properties converge, the Tibetan Plateau or the Indo-Gangetic plain. The transported air masses subsequently mix in the ASMA.

Figure 1c illustrates transport from the Pacific to the measurement locations in relation to three typhoons (named Nepartak, Nida and Omais) that occurred during the relevant time period (we use the typhoon best track data of the Japan Meteorological



Agency). The typhoons uplift a large number of air parcels from the maritime surface into the eastern edge of the ASMA. Depending on the position of the ASMA modes and the typhoons, the uplifted air masses are circulating in the outer edge

of the ASMA (e.g. for Nida and Nepartak) or they circle mostly in the eastern mode, and the inner area of the ASMA (e.g. Omais). Because of the multiple circulations in the ASMA the typhoons influence the measurements with a delay in time of several days. The impact of typhoons on the air masses in the ASMA and the ATAL has been reported before (e.g. Li et al., 2020; Hanumanthu et al., 2020).

Figure 1d presents a transport pathway from the Arabian Sea to the ASMA. The transport of the air parcels occurs in

four steps. At first, the air leaves the MB into the free troposphere over the Arabian see, possibly due to shallow, maritime convection. Secondly the air is transported eastward within the free troposphere to the foothills of the Himalaya or to the Bay of Bengal. During this transport, over the Indo-Gangetic plain, or at the Bay of Bengal, deep convection uplifts the air to the southern edge of the ASMA, which is the third step of transport. In the last step, the air masses circle in the ASMA until they are measured at Nainital. A similar long-range transport pathway to the Himalayas can be observed for the Bay of Bengal.

However from the Bay of Bengal more air parcels can enter the ASMA directly from the maritime boundary than from the Arabian Sea. In Appendix B, the transport pathways from the Bay of Bengal, South India, South-East-Asia and the remaining ocean and continent are presented as well.





(a) Asian Highlands/Tibetan Plateau

(b) Indo-Gangetic plain

(c) Tropical Western Pacific

(d) Arabian See

**Figure 1.** Examples of backward trajectories of air parcels of all of the 15 measurement days from the ATAL measurement to the MB, categorised by the source region. Shown are trajectories of scenario E5-kin-C. Colors indicate the time when the air parcels left the MB. Gray dots at the bottom show the horizontal position of the APs 48 h before they crossed the MB from below. In (c), additionally, tracks of three typhoons are plotted (Nepartak, Nida, Omais), where each point is colored like the trajectories and the mean time of occurrence.



### 3.2 Scenario intercomparison of contributions from source regions and related transport pathways

Although the general transport pathways from the MB to the measurement locations, as presented in Sect. 3.1, exist in all
simulation scenarios with the Lagrangian transport models, the contributions of different source regions can differ, depending
on the used scenario. Here, we present the differences and similarities of the vertical and horizontal distribution of the source
regions between the different scenarios. For the analysis, we use all 15 measurement days and 90-day backward trajectories.

First, the fraction of air from different atmospheric layers is calculated for all model scenarios (see Fig. 2). Wide agreement
can be found with ERA5 even when models, integration step-sizes, and vertical velocities are varied. The total amount of
transport from the MB lies between 74% and 80% for the ERA5 scenarios. The distribution from the LT, UT, or LS shows only
some minor differences. Large disagreement is found between the diabatic and kinematic vertical velocities using ERA-Interim.
With kinematic vertical velocities, only around 60% of the air parcels originate in the MB, while with diabatic velocities the
results are closer to the ERA5 scenarios (around 75% amount of transport from the MB). Disagreement in ERA-Interim, when
varying the vertical velocities, was also found in other studies (e.g. Ploeger et al., 2010; Li et al., 2020; Legras and Bucci,
2020). The low resolution ERA5 data set maintains the higher consistency of ERA5 in comparison to ERA-Interim and the
total transport from the MB is only slightly reduced, probably because the vertical resolution is unchanged and higher vertical
velocities over the continent remain also higher in ERA5lr as well. A large increase of transport from the MB is caused by the
onset of the convection parameterisation in scenario E5-kin-M-ECP. In this case only around 10% of the air parcels originate
outside the MB and no air parcels originate in the LT. The latter can be explained, when following the air parcels backward in
time: If the air parcels enter the LT during backward calculations they likely enter a region where the ECP is triggered. Then
transport into the MB takes place immediately.

Second, the fraction of air from different model boundary regions contributing to the ATAL is compared for all model
scenarios (see Fig. 3b). Model scenarios driven with ERA5 show very similar results. For ERA5 scenarios, about 40% of the
air parcels that originate from the MB, originate from the Tibetan Plateau and other Asian highlands. Around 5% to 10% of
air parcels come each from South India and South East Asia. The contributions from the Indo-Gangetic plain is between 10%
and 20%, with higher values for diabatic calculations. The Indian subcontinent, i.e. the Indo-Gangetic plain and South India
together have the largest contribution from the continent. Around 25% of the air parcels that come from the MB originate
from oceans, mostly the Western Pacific and many are related to tropical cyclones. In contrast to ERA5, using ERA-Interim
increases the contribution of air parcels from the oceans up to around 40%. This increase does not rely on the variation of
the vertical velocity and is robust for all scenarios driven with ERA-Interim. The contribution from the Indo-Gangetic plain
is reduced to 5%-10% with ERA-Interim compared to ERA5, and shows also a difference between diabatic and kinematic
velocities. With the extreme convection scenario the contribution from the oceans is smaller than in the cases with ERA5.
However, with the ECP the contribution from the Indo-Gangetic plain and the foothills of the Himalayas increases strongly
at the expense of contributions from the Tibetan plateau and the oceans. Finally, the ERA5 low resolution scenarios show a
decrease of consistency between the diabatic and kinematic approach. The kinematic approach has a bias to more transport





from the ocean in the low-resolution data in comparison to the fully resolved ERA5 data, while the results for the diabatic approach show only minor difference to the fully resolved ERA5 data in the statistics.

With the ensemble scenario (E5-kin-M-Diff), we tested uncertainties due to unresolved sub-grid scale wind fluctuations and found standard deviations lower than 1% for the distribution of air parcels to different vertical layers or horizontal regions.

Therefore, the sub-grid scale wind fluctuations do not cause a serious bias in this analysis. The ensemble scenario E5-kin-M-1800s was used to investigate possible biases between the models due to different time-steps. However, as the results remain almost unaltered for MPTRAC when the time-step is varied from 180 s to 1800 s we rule out a serious bias for our analysis.

The probability density function (PDF) for air parcels leaving the model boundary is shown in Fig. 4. All scenarios except the ECP scenario show a very similar pattern, with most dominant transport from a region centered at the eastern foothills of

the Himalayas and the south-eastern Tibetan plateau (around 50%), while transport from other regions is much less but not irrelevant. If contours for ERA5 scenarios and ERA-Interim scenarios are compared, air parcels are dispersed more with ERA-Interim, particularly in direction of the ocean, while ERA5 resolves more transport at the continent and disperses the air parcels less. Using the scenario with ECP deforms the pattern even more, because more transport happens then at the Indo-Gangetic plain, due to higher convective available potential energy (CAPE) in this region than at the Tibetan plateau. The results of the

ECP depend on the selection of the CIN and CAPE thresholds. We used the CIN threshold to remove spurious parameterized convection over the Persian Golf (see Appendix F).

In summary, ERA5 provides improved robustness against changes of the vertical velocity between the kinematic and diabatic approach in comparison to ERA-Interim, and yields very good agreement between the two Lagrangian models. Using the ECP in MPTRAC indicates that even scenarios with ERA5 could miss effects of unresolved convection, particularly locally over

the Indo-Gangetic plain. This leads to difficulties when distinguishing the contributions from the Tibetan plateau and the Indo-Gangetic plain.




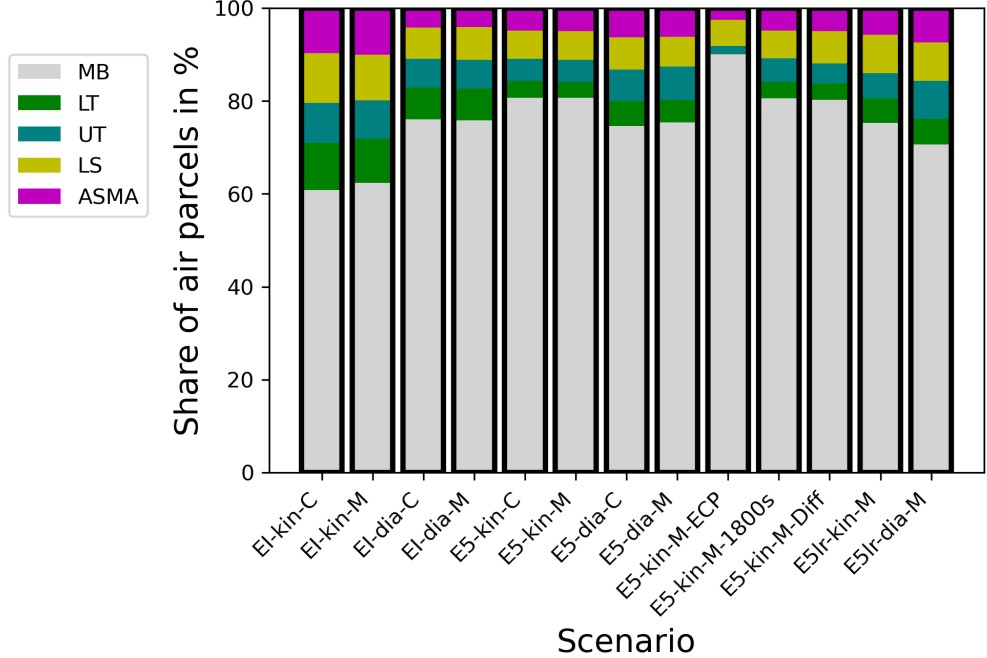

**Figure 2.** Vertical classification of air parcel origin after 90 days of backward trajectory calculations with different scenarios.



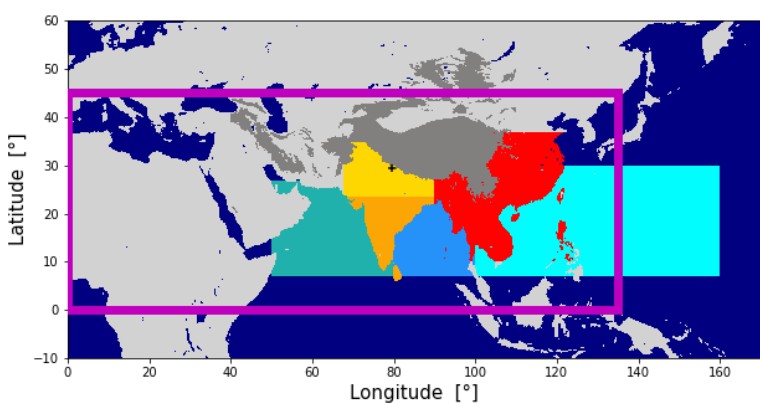

(a) Definition of the Regions

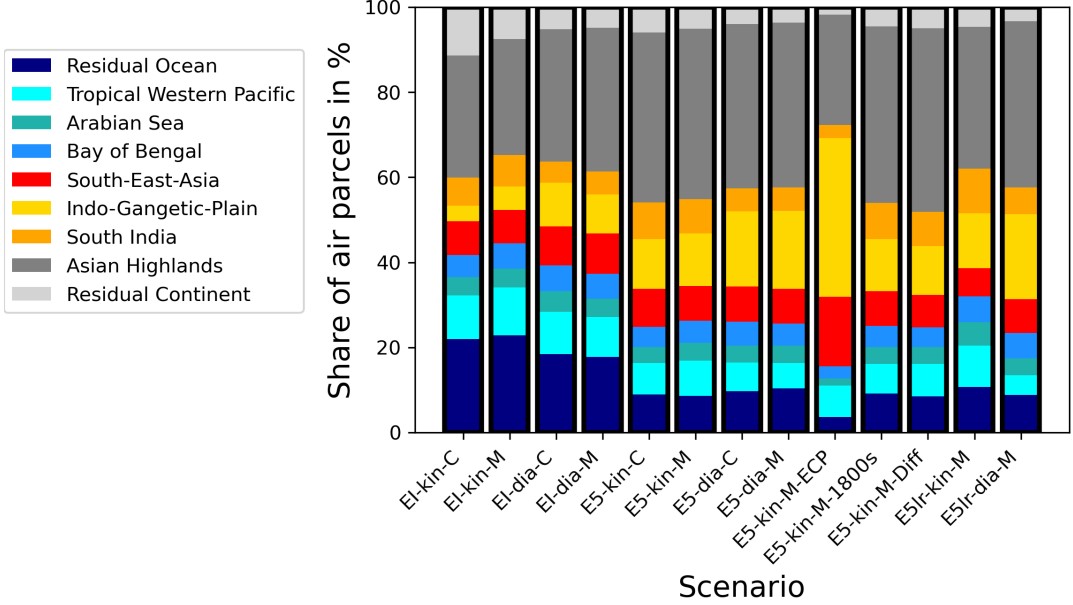

(b) Classification after 90 days

**Figure 3.** (a) shows the definition of contributing regions. The purple box marks an area that contains most of the air parcel that circulate in the ASMA. (b) shows the horizontal classification of the air parcels according to the surface regions after 90 days backward trajectory time. Shown is the fraction of air parcels that reach the MB.



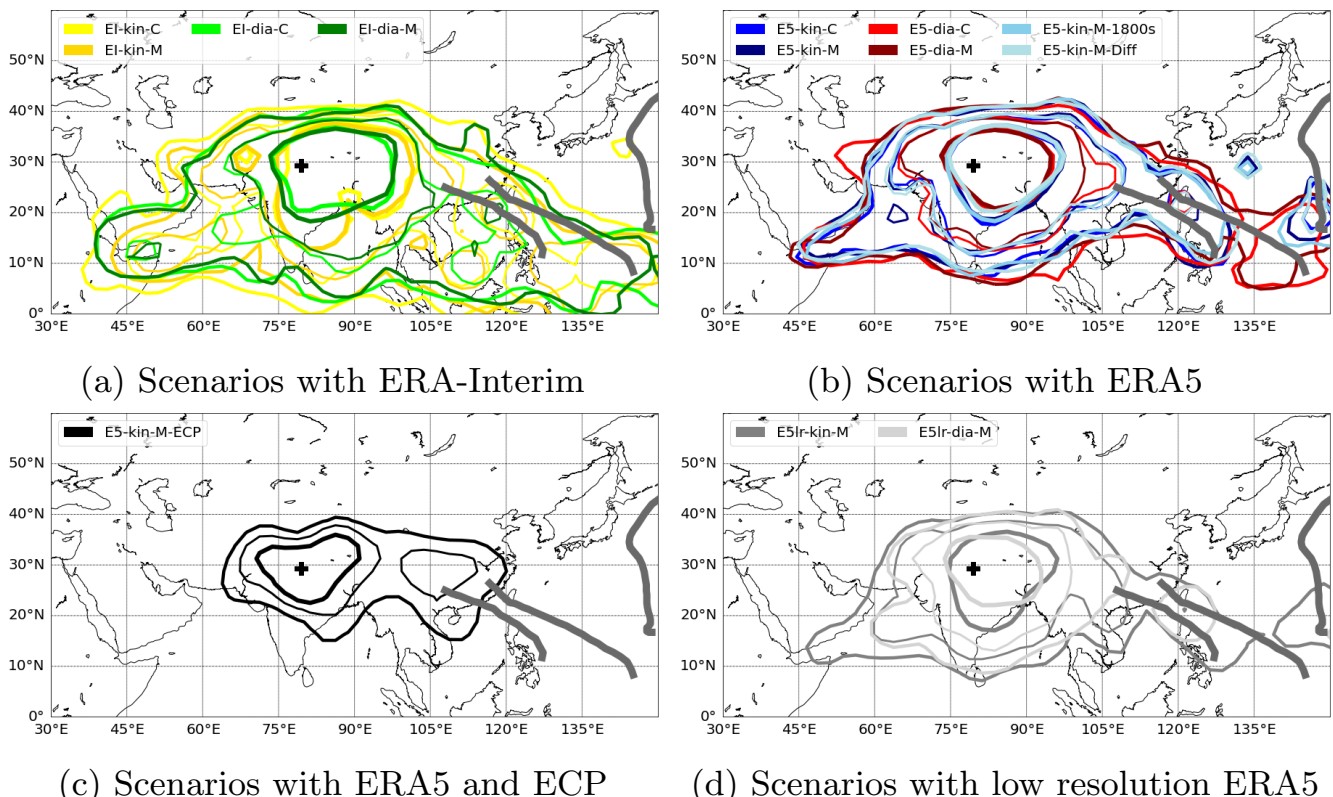

**Figure 4.** Contours of the probability density function (PDF) of the surface sources for 90 days backward trajectories and all measurements. Colors indicate the different scenarios. For each scenario the inner, thick contour encloses 50% of the points where air parcels left the MB and the outer contour encloses 90% of them. In between, a thin contour is shown for 75%. For the sake of clarity, the contours are smoothed by a Gaussian kernel. The black cross indicates the position of Nainital. The three thick gray lines show typhoon tracks.

### 3.3 Scenario intercomparison of the temporal evolution of transport from the MB to the measured ATAL

The transport time of the air parcels from the MB to the ATAL affects aerosol formation. The time of the transport is therefore an important parameter to analyse. An analysis of the temporal evolution of the transport process from the MB to the measurements also highlights some differences between the model scenarios.

To emphasize the possible lifetimes of air masses transported to the location of the measurements in the UTLS, Fig. 5a shows the frequency of air parcels leaving the MB per day at different times before the measurements for the different scenarios. These frequencies are classified in two categories, the continents and the oceans. Most of the relevant maritime processes that transport air masses out of the MB into the upper atmosphere take place more than two weeks before the measurements. This can be found for all scenarios except for the scenario with ECP. This scenario shows very rapid transport out of the MB three





days before the measurements. For scenarios with ERA-Interim the frequency of air parcels leaving the MB is higher than for ERA5 if transport times longer than 40 days are considered, in particular for the diabatic scenarios with ERA-Interim.

Considering transport from the continent, for ERA-Interim only a few air parcels originate from the continent with transport times less than two weeks, independent of the used vertical velocity (diabatic or kinematic). This resembles the results of
Hanumanthu et al. (2020), who used diabatic CLaMS trajectories driven by ERA-Interim. In contrast, all scenarios with ERA5 show that air from the continent can be transported much faster to the location of the measurements even in much less than two weeks. This is likely due to better resolved convection in ERA5 in comparison to ERA-Interim. This fast transport at the beginning is maintained with the low-resolution ERA5 data, although it is reduced in temporal and spatial resolution in comparison to the full ERA5. Furthermore, using the ECP, reveals that also ERA5 potentially underestimates convection in
the first days. The fast transport with ECP is caused by the frequent triggering of the parameterisation over the continent, particularly over the Indo-Gangetic plain, where the atmosphere often shows convective parameters (CAPE, CIN) that suggest unstable conditions. However, our approach of the ECP has to be considered as an upper limit for convective transport. In summary, in ERA5 air masses are transported from the continental MB to the ATAL relatively fast (within two weeks), so that less air parcels remain to originate from the maritime MB, while with ERA-Interim this effect reverse. With ERA-Interim only
few air parcels are transported fast to the ATAL from the continent, while more and older air parcels originate from the oceans.

Figure 5b shows the accumulation of air parcels within the ASMA during the transport processes to the ATAL over Nainital. The differences between the scenarios can be understood when the transport is described following the calculations backward in time, i.e. when we look at the "draining" of the ASMA during backward calculations. Our calculations show that in all scenarios most of the air parcels start in the ASMA. When the air parcels are traced back in time, the share of air parcels in
the ASMA for scenarios with ERA5 and ERA-Interim starts to diverge. In the first two weeks backward in time (-14 days to 0 days), scenarios with ERA5 show more transport back into the MB than scenarios with ERA-Interim, i.e. the share of air parcels in the MB increases faster with ERA5 and the share in the ASMA declines faster than with ERA-Interim. This is in agreement with faster transport from the continent with ERA5 than with ERA-Interim as described before. After calculating the trajectories further back in time (longer than two weeks), the share of air parcels in the ASMA starts to converge again for
the ERA-Interim scenarios with diabatic velocities (EI-dia-C, EI-dia-M) and the ERA5 scenarios. This convergence is partly caused by the increased backward transport to the maritime MB in the scenarios with ERA-Interim and the diabatic scheme, while with ERA5 the transport to the maritime MB is smaller in comparison (see also Fig. 5a, left panel, -70 days to -30 days). The ERA-Interim scenarios with kinematic approach (EI-kin-C, EI-kin-M) diverge further from all other scenarios, showing a much lower share of air parcels in the MB than other scenarios and the most air parcels in the UTLS. After around two
weeks the share of air parcels in the UTLS increases more in the scenarios with ERA-Interim and kinematic vertical velocities (see lower plot in Fig. 5b) than in all other scenarios. Ploeger et al. (2010, 2011) demonstrated more dispersion in backward trajectory calculations with ERA-Interim and the kinematic approach than with the diabatic approach. This effect explains the large difference of the scenarios EI-kin-C and EI-kin-M from the other scenarios and why many air parcels are transported back into the UTLS with these scenarios.



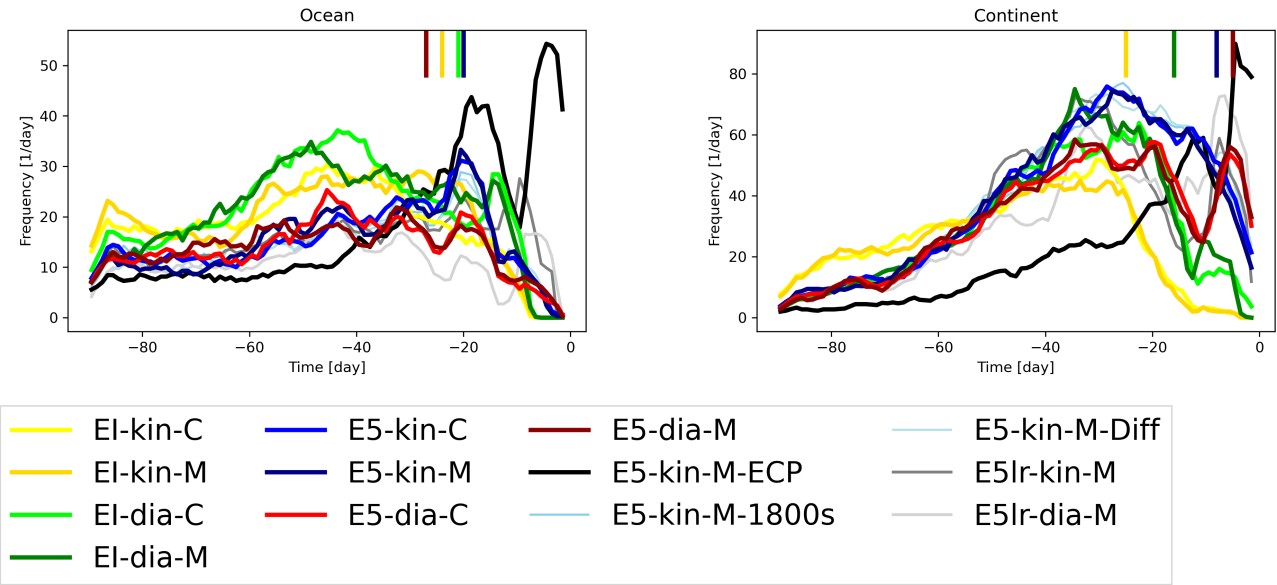

(a) Horizontal Classification

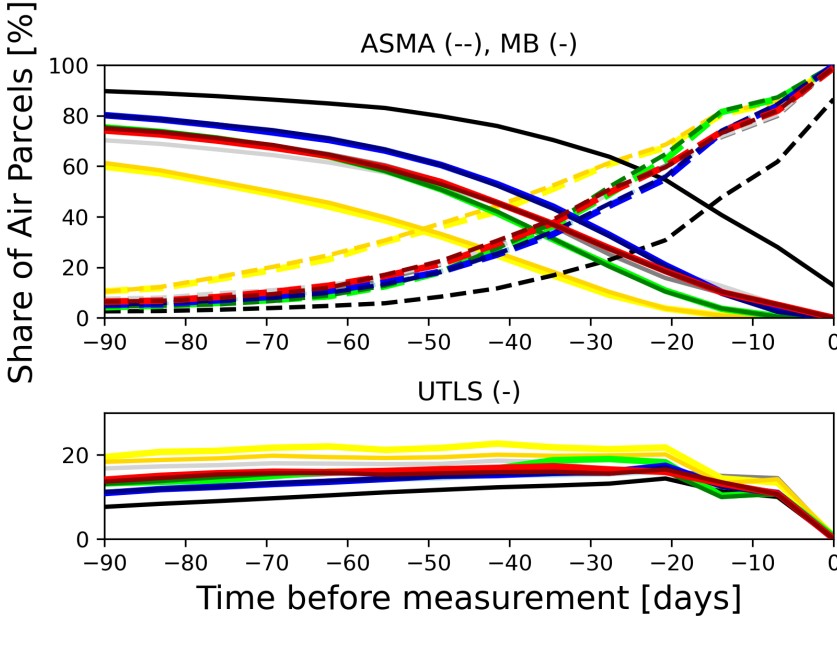

(b) Vertical Classification

**Figure 5.** Time evolution of transport from the MB to the ASMA within 90 days, relative to the start of the trajectories. (a) shows the frequency of air parcels leaving the MB smoothed with a one-week running mean. The short lines at the top indicate for some scenarios the maximum time for the first 300 air parcels with the smallest transport time. (b) shows the share of air parcels that are within the ASMA (upper panel, dashed line), the share of air parcels that are in the MB (upper panel, solid line) and the share of air parcels that are in the UTLS (lower panel, solid line).





### 3.4 Backscatter changes associated with changes in the transport and source regions

Hanumanthu et al. (2020) found a distinct day-to-day variability of the ATAL backscatter intensity. This variability may be correlated with daily to weekly changes of the transport within the highly variable anticyclone and variability of tropical convection and therefore with changing surface source regions. To analyse the changes from measurement to measurement, the contribution of different source regions to the vertical ATAL profile has been reconstructed for every balloon flight separately. Subsequently, the relative deviation of the contribution of a specific region on one day from the mean contribution during all measurements normalized by the mean contribution was calculated as a relative, normalized deviation (in percent). Accordingly, $RND(t) = \left(\frac{C(t)}{\overline{C(t)}} - 1\right) \times 100$ was calculated for every day of a selected scenario. $RND(t)$ denotes the relative, normalized deviation for measurement day $t$ and $C(t)$ denotes the contribution of air parcels for the measurement day $t$ from the selected region. The contribution $C(t)$ is measured as the ratio between the number of air parcels from the selected region to the total number of air parcels for the measurement day. $\overline{C(t)}$ is the time averaged contribution over all measurement days for the selected scenario. This calculation has been done for all scenarios separately to allow a direct day-to-day intercomparison of the scenarios.

Figure 6 shows the normalized deviations for the Indo-Gangetic plain and the Western Pacific Ocean for 13 measurement days. The days with no ATAL or with large cirrus cloud coverage have been excluded from the analysis (12 and 15 August). Furthermore, we focus on the Indo-Gangetic plain and the Western Pacific Ocean as they show the most robust and significant results in comparison to other source regions.

Figures 6a,b show the relative, normalized deviation for every measurement day. Overall, all scenarios indicate a clear temporal evolution of the contribution from the two regions during the weeks of the campaign in August: The scenarios show that in the early phase of the campaign (2, 3, 6 and 8 August) the contributions from the Indo-Gangetic plain were enhanced relative to other days, while in the later phase (19-30 August) contributions from the Indo-Gangetic plain were relatively low. For contributions from the Western Pacific the opposite is found, because of increased impact of typhoons on the measurements at the end of August (see Fig. 6a,b). The variability from day-to-day, in contrast to the general temporal evolution, has a magnitude of several 10%. For some measurement days, the scenarios show similar day-to-day differences (e.g. 2-8 August for the Tropical Western Pacific), but for other periods the day-to-day variability differ strongly between scenarios (e.g. 23-30 August for the Tropical Western Pacific).

Figures 6c and 6d show the relation between the averaged measured BSR for every day and the daily relative, normalized deviation. For the Indo-Gangetic plain in all scenarios large backscatter ratios coincide with large normalized deviations, although in some scenarios this relation is considerable weak. For the Tropical Western Pacific, in all scenarios low backscatter ratios clearly coincide with higher normalized deviations.





(a) Indo-Gangetic plain

(b) Tropical Western Pacific

(c) Indo-Gangetic plain

(d) Tropical Western Pacific

**Figure 6.** (a) and (b) show the relative, normalized deviation for each measurement day of the Indo-Gangetic plain and the Tropical Western Pacific. (c) and (d) show the relations between the daily averaged backscatter intensity and the relative, normalized deviation. The lines show linear fits for each scenario. Colored dots show results for all scenarios. The colorcode for the dots is the same as in Fig. 5, where every color correspondence to one scenario.



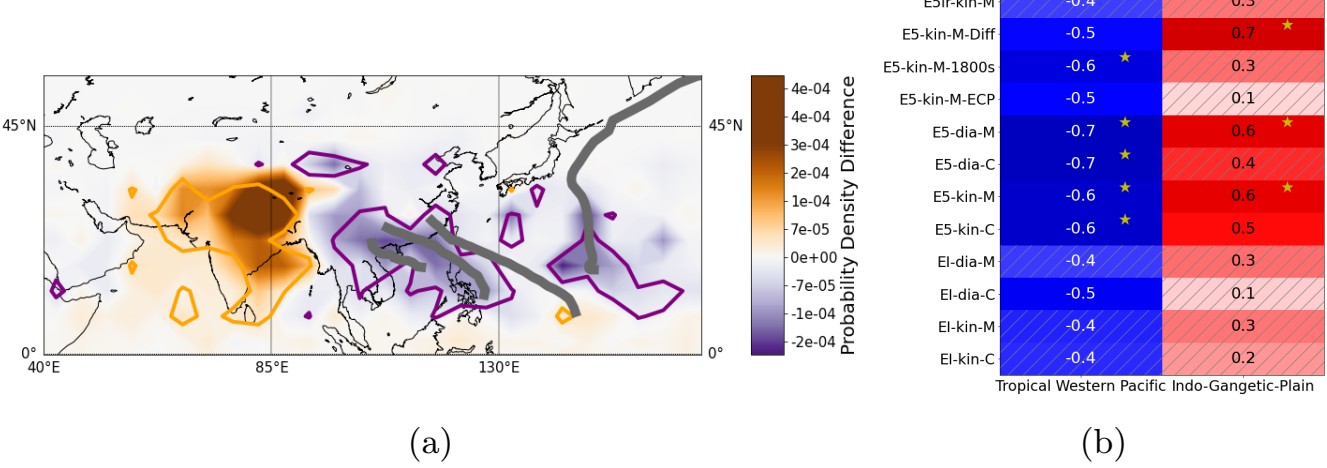

(a)                                             (b)

**Figure 7.** (a) Difference between the source region PDF of the five days with maximum ATAL backscatter intensity and five days with minimum backscatter intensity, derived from the multi scenario mean. The orange contours show areas where at least two thirds of the scenarios indicate high values during a strong ATAL. The purple contours show the same for the weak ATAL cases. Data is given on a $5°$ grid. The gray lines indicate the tracks of tropical storms that influenced the measurements. (b) Spearman correlation coefficient for the relation between the daily BSR and the contributions of different regions for different scenarios. Crossed out areas hatch insignificant results (p-value higher 0.10). Boxes with yellow stars indicate p-values lower than 0.05. Colors emphasize positive (red) and negative (blue) correlations.

375    To further compare transport of those days with a low ATAL backscatter intensity above Nainital with those days with a high one, we also calculated the PDFs of surface source regions for each day separately. In particular, we selected the five days with strongest (3, 6, 26, 17, 19 August) and the five days with the weakest backscatter intensity (21, 23, 30, 18, 5 August) for the analysis. Figure 7a shows the differences between the two PDFs for the multi-scenario mean. The results indicate that transport from the Indo-Gangetic plain, India, the Bay of Bengal and the northern part of the Tibetan Plateau may have strengthened the

380    ATALs backscatter intensity during the measurements. The intensity of the ATAL over Nainital again seems to be low when more transport from the tropical cyclones and the region of their landfall occurred. Although the PDF differences vary strongly from scenario to scenario (see Fig. D1 in the Appendix), at least two third of the scenarios agree on these characteristics.

For a complete analysis, we used the Spearman rank-order correlation coefficients to check if there is a monotonic relation between the measured backscatter and the contribution of specific regions. Figure 7b summarizes these correlations for the

385    Indo-Gangetic plain and the area of active typhoons at the Tropical Western Pacific. Correlations to other regions can be found in the Appendix in Fig. E1.

We find a significant (p<0.1) and robust negative correlation between the backscatter intensity of the ATAL and the Western Pacific influenced by typhoons. This correlation remains to be present both for the simulation with ECP with maximum strength of convection and with ERA-Interim where in contrast convection is underestimated. Most scenarios also indicate a positive





correlation between the backscatter intensity of the ATAL and enhanced contributions from the Indo-Gangetic plain. Although,
impacts of unresolved convection likely can be neglected for the relation to the West Pacific, the correlation for the Indo-
Gangetic plain changes substantially from 0.6 to 0.1 with parameterized convection (see E5-kin-M-ECP). Moreover, for the
Indo-Gangetic-plain the correlation in the scenarios E5lr-dia-M and EI-dia-C remains insignificant. The most significant results
however, are obtained in the ERA5 scenarios, supporting the hypothesis that polluted air from the Indo-Gangetic plain led to
higher backscatter intensity and vice versa clean maritime air from the Western Pacific lead to a dilution of the ATAL and
therefore to a weaker BSR intensity.

Most scenarios furthermore have a positive correlation for contributions from the Arabian Sea, the rest of India, and the
Bay of Bengal. Such relations seem plausible given the transport pathways from the Arabian Sea and the Bay of Bengal to
the ATAL over Nainital, which include a period of horizontal transport in the polluted troposphere over India before a second
step of upward transport in deep convection. Additionally, air masses from the Arabian Sea could carry dust from the Arabian
Peninsula. Air from South-East-Asia is also weakly correlated with a decrease of the ATAL backscatter intensity, which is
possibly related to the landfall of some typhoons.

Other source regions have been considered to establish such relations, but given the limited amount of data and systematic
model uncertainties, no further robust results were found. The positive relation between the Tibetan plateau and the backscatter
intensity of the ATAL, found by Hanumanthu et al. (2020) was reproduced for similar scenario set-ups (see EI-dia-C). However,
some scenarios show even negative correlations.

Our results corroborate that the transport calculations presented here are capable of capturing the general evolution and
patterns during the course of August robustly, but might differ if day-to-day changes are considered. The observed general
temporal evolution might be related to a large scale change of the meteorological conditions during August, that led to a
shift to more air masses from the Western Pacific and prior typhoons and less transport from the Indian Subcontinent. The
differences in the representation of day-to-day changes might arise from the uncertainty about the daily changing convection.

## 4 Discussion and conclusions

In this study, we investigated the source regions of the ATAL in August 2016. To identify the source regions at the model
boundary and the transport pathways contributing to the ATAL and to investigate the sensitivity of the applied methods, dif-
ferent trajectory calculations were conducted. Simulations with different model scenarios using different Lagrangian transport
models (CLaMS and MPTRAC), wind data (ERA-Interim and ERA5), vertical velocities (kinematic and diabatic), integration
time-steps and a convection parameterisation (ECP) were analyzed. Additionally, we correlated daily contributions of source
regions at the surface to daily measured COBALD backscatter intensities at ATAL altitudes to quantify the role of the regions
for the intensity of the ATAL.

Most of the air from the MB that influenced the measurements originated at the Tibetan Plateau (i.e. $\approx$ 30%-40% of air
masses originating at the MB). This is found for all scenarios, except for the scenario with the extreme convection parame-
terisation. In the scenario with the ECP (E5-kin-M-ECP), the Indo-Gangetic plain is contributing most ($\approx$ 30%). The Indo-





Gangetic plain is the second largest continental contributor (10 – 20 %) to the air masses influencing the measurements in all other scenarios (except for ERA-Interim with a diabatic approach). The contribution from the Indo-Gangetic plain, however, is much smaller than from the Tibetan Plateau in those scenarios. In summary, most of the upward transport takes place at the eastern part of the Indo-Gangetic plain that extends to the Bay of Bengal, including the foothills of the Himalayas and the Tibetan Plateau. These regions have been found to be dominant for the transport into the ASMA and the ATAL also by other studies (e.g. Bergman et al., 2012; Bucci et al., 2020; Hanumanthu et al., 2020).

In our simulations, a small amount of air was transported from South-East-Asia and North India, to the ATAL as well. Such transport processes contributing to the ATAL have also been reported before (e.g. Vernier et al., 2018; Bucci et al., 2020; Zhang et al., 2019a, b, 2020). Air masses from the maritime boundary layer were transported to the measurement locations in significant numbers as well. This includes mostly air masses from surrounding seas, such as the Arabian Sea, the Bay of Bengal and the Western Pacific. Typhoons in the Tropical Western Pacific played an important role for the transport process from the maritime boundary layer, which is in good agreement with previous studies, which showed their relevance for the UTLS and the composition of the ATAL (Li et al., 2017, 2020; Hanumanthu et al., 2020).

By studying the transport pathways and times, we showed some systematic differences between simulation scenarios that are related to the representation of convection and the diabatic ascent in the ASMA. ERA5 has a better representation of convective updrafts and tropical cyclones compared to the ERA-Interim reanalysis, attributed to its better spatial and temporal resolution and other improvements of the ECMWF forecast model and data assimilation scheme. Therefore, the fraction of air from the MB transported upward to ATAL altitudes is lower or about equal in scenarios with ERA-Interim in comparison to scenarios with ERA5. This is in particular true over the continent. Hence, in ERA-Interim convection over the continent is less frequent than in ERA5, so that larger fractions of air parcels originate from remote maritime regions with ERA-Interim (40% vs. 23% of all air parcels from the MB).

ERA-Interim simulations show also strong differences with regard to the transport from the MB into the UTLS, when the vertical velocity is varied between diabatic velocities (75% of all air parcels) and kinematic velocities (60% of all air parcels). These differences between kinematic and diabatic trajectories are strongly reduced, when ERA5 is used, where the diabatic approach shows similar fractions of air transported from the MB to ATAL altitudes like the kinematic approach (74% vs. 80%). Large differences with regard to the vertical transport in typhoons between ERA5 and ERA-Interim have been reported before by Li et al. (2020) and an improvement of consistency between vertical velocities by Legras and Bucci (2020) in the Asian monsoon region.

Although ERA5 resolves convection better than ERA-Interim (Hoffmann et al., 2019), it might still underestimates the extent of fast vertical transport caused by deep convection at the foothills of the Himalayas and at the Indo-Gangetic plain. This possible deficiency is indicated by the simulation scenario with ECP, that shows a strong increase of convection near Nainital at the Indo-Gangetic plain and the foothills. Our results show, that ERA5 provides a significant improvement for the simulation of transport processes in the Asian monsoon region with regard to the consistency between scenarios with different models and vertical velocity schemes. However, ERA5 possibly still has limitations with regard to the representation of the convection, which needs to be evaluated in further studies that take observations of convection into account. In addition, our





study shows for the employed Lagrangian model (MPTRAC and CLaMS) minor differences. These differences are likely caused by differences in the integration scheme and the interpolation method. Both models are equally valuable in case of the
present analysis.

Taking into account the measured backscatter intensity of the ATAL, we found two regions with a significant and robust impact on the ATALs variability. Meteorological conditions that are favourable to transport from the Indo-Gangetic plain increase the ATAL backscatter intensity, while conditions favourable to transport of air masses from the Tropical Western Pacific and the influence of typhoons decrease the ATAL backscatter intensity. In case of the Tropical Western Pacific, these
findings hold for the different sensitivity calculations carried out, and hence we corroborate the results by Hanumanthu et al. (2020), by showing that this correlation is robust despite the systematic uncertainties represented by the different simulation scenarios. In case of the Indo-Gangetic plain 10 of 13 scenarios are underpinning this finding of a positive correlation, while the remaining three scenarios show very low and insignificant correlations. To completely remove the remaining uncertainties, further observations in the region are needed.

Our findings are in agreement with results of previous studies. Studies found ammonium nitrate particles as a major component of the ATAL (Höpfner et al., 2019). Ammonia, the precursor of this aerosol is emitted frequently over the Indo-Gangetic plain, which is an area of active agriculture and industry (Kuttippurath et al., 2020) and could be transported fast enough into the ASMA within hours to a few weeks according to our calculations. Transport within typhoons up into the UTLS can provide clean and dry air from the ocean (Li et al., 2020) leading to a reduction of the backscatter intensity of the ATAL, as also shown
in our simulations. The possible role of dust from the Asian deserts or highlands for the formation of the ATAL is discussed in the literature (e.g. Vernier et al., 2011; Bossolasco et al., 2021). Our calculations do not disprove that dust from the Tibetan Plateau could have contributed to the ATAL in August 2016, but indicate that dust is likely not essential to understand the observed variability of the ATAL backscatter intensity during this period. Indeed, the simulations indicate a large potential for transport of dust into the ASMA from the Asian Highlands, i.e. mostly the Tibetan Plateau, but this transport more likely leads
to a constant background over the observed period.

*Code and data availability.* The ERA-Interim and ERA5 reanalysis data are available from ECMWF. The CLaMS code can be accessed from the Jülich GitLab server: https://jugit.fz-juelich.de/clams/CLaMS (last access: 1 December 2022). MPTRAC can be accessed under the terms of the GNU General Public License at the GitHub repository: https://github.com/slcs-jsc/mptrac (last access: 1 December 2022). We made use of the Japanese best track data of typhoons for the analysis of the relation between tropical storms and the transport pathways
(https://www.jma.go.jp/jma/jma-eng/jma-center/rsmc-hp-pub-eg/besttrack.html, last access: 15 July 2022). Balloon sounding data will be provided on request by S. Fadnavis (suvarna@tropmet.res.in).



## Appendix A: Definition of source regions

To construct our surface source regions, first the geopotential is considered to define the Asian Highlands as the region in Asia with a geopotential larger than $15000 \ \mathrm{m^2 \ s^{-2}}$ (corresponding to a geopotential height of approximately $1.5 \ \mathrm{km}$) similar to Hanumanthu et al. (2020). Secondly, a land-sea mask is used to distinguish between oceans and continents. The continental regions are defined by the boxes found in Table A1. The boxes defined in Table A2 define maritime regions, where only those regions are included that are part of the sea according to the land-sea mask.

**Table A1.** The continental source regions are defined by the overlap of longitudinal and latitudinal restricted boxes and the continent without the Asian Highlands. The Asian Highlands are defined by a GPH criteria.

| Name | Label | Minimum Longitude | Maximum Longitude | Minimum Latitude | Maximum Latitude |
|---|---|---|---|---|---|
| Asian Highlands | AH | 40° E | 110° E | 20° N | 90° N |
| South India | SI | 65° E | 90° E | 5° N | 23.5° N |
| Indo-Gangetic plain | IGP | 67.5° E | 90° E | 23.5° N | 35° N |
| South-East-Asia | SEA | 100° E | 160° E | 7° N | 30° N |

**Table A2.** The maritime source regions are defined by the overlap of longitudinal and latitudinal restricted boxes with the seas.

| Name | Label | Minimum Longitude | Maximum Longitude | Minimum Latitude | Maximum Latitude |
|---|---|---|---|---|---|
| Arabian Sea | AS | 50° E | 80° E | 7° N | 27° N |
| Bay of Bengal | BOB | 80° E | 100° E | 7° N | 27° N |
| Tropical Western Pacific | TWP | 100° E | 160° E | 7° N | 30° N |




## Appendix B: Transport pathways from source regions

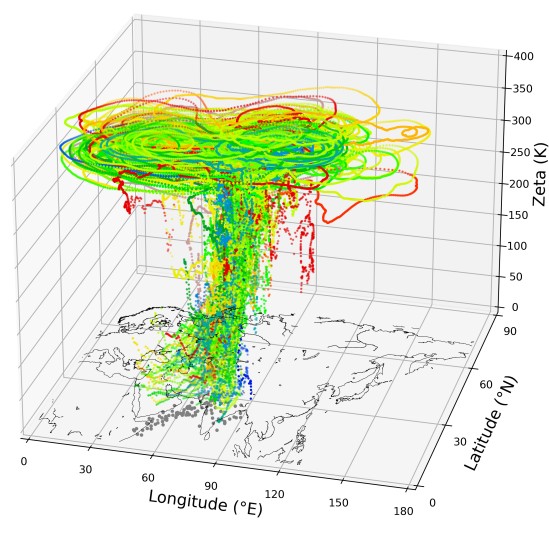

(a) South India

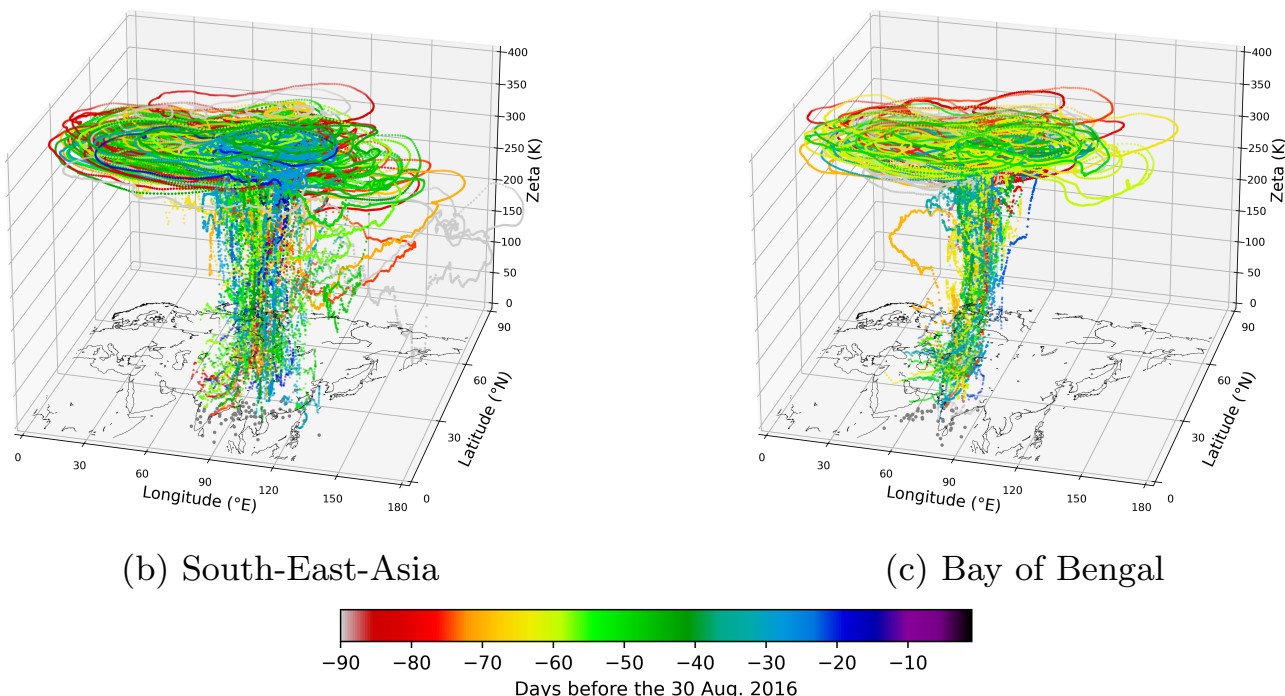

(b) South-East-Asia    (c) Bay of Bengal

**Figure B1.** Exemplary backward trajectories of air parcels of all of the 15 measurement days from the ATAL measurement to the MB, categorised by the source region. Colors are chosen as in Fig. 1.





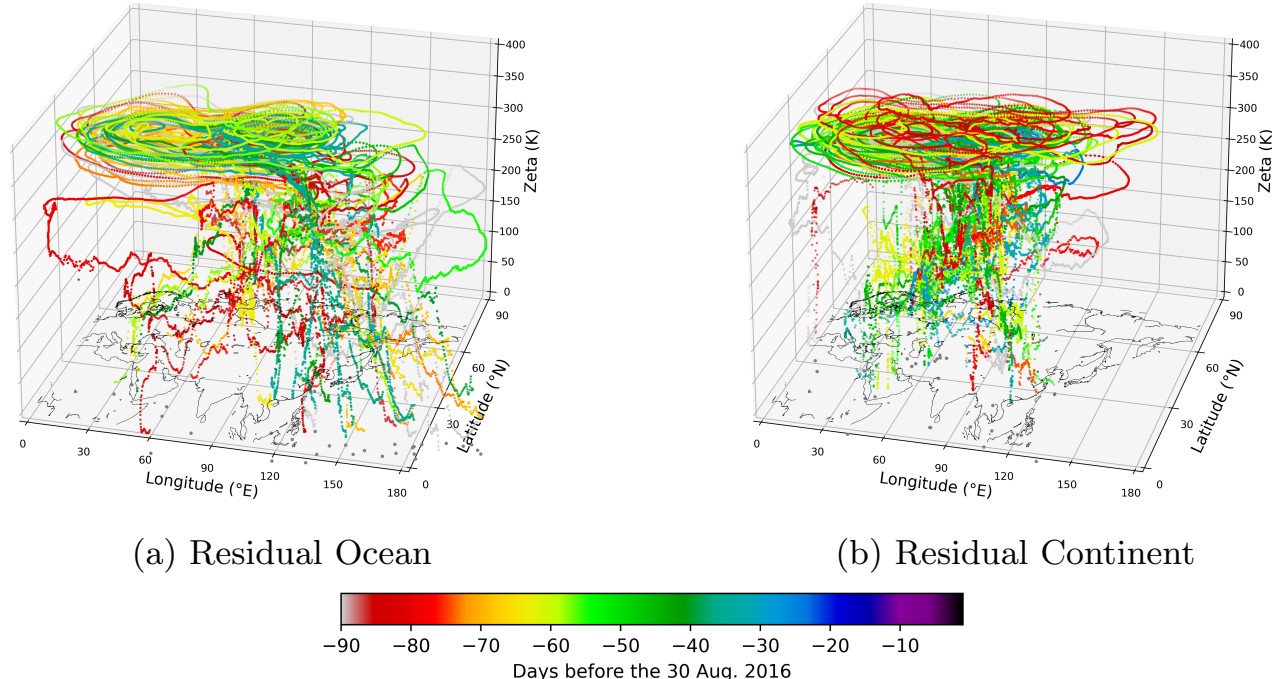

**Figure B2.** Exemplary backward trajectories of air parcels of all of the 15 measurement days from the ATAL measurement to the MB, categorised by the source region. Colors indicate the time when the air parcels leaves the MB. Gray dots at the bottom show the horizontal position of the APs 48 h before they cross the MB from below.



## Appendix C: PDF for different Scenarios

(a) EI-kin-C

(b) EI-dia-C

(c) E5-kin-C

(d) E5-dia-C

(e) E5-kin-M

(f) E5-kin-M-ECP

**Figure C1.** PDFs for the MB source region for some scenarios.



**Appendix D: PDF differences for different Scenarios**

(a) EI-kin-C

(b) EI-dia-C

(c) E5-kin-C

(d) E5-dia-C

(e) E5-kin-M

(f) E5-kin-M-ECP

**Figure D1.** PDF differences for days with high and low ATAL backscatter for specific scenarios, similar to Fig. 7a, illustrating the sensitivity of the PDF differences to the chosen scenario.



**Appendix E: Correlation to all regions**



**Figure E1.** All Spearman correlation coefficients for the relation between the daily BSR and the contributions of different regions for different scenarios. Crossed out areas hatch insignificant results (p-value higher 0.10). Boxes with yellow stars indicate p-values lower than 0.05. The regions are ordered according to the scenario mean from lowest to highest correlations.





**Table E1.** Mean transport time from the MB into the UTLS in days, determined by the difference between the leaving time at the MB and the arrival time above 340 K. Abbreviations are as defined in Table A1 and A2.

| Scenario | All | Res. Oc. | TWP | AS | SEA | BOB | IGP | SI | AH | Res Cont |
|----------|-----|----------|-----|-----|-----|-----|-----|-----|-----|----------|
| EI-kin-C | 7.2 | 13.8 | 9.1 | 5.9 | 2.2 | 6.2 | 7.0 | 9.8 | 5.6 | 6.0 |
| EI-kin-M | 7.2 | 13.0 | 12.7 | 5.4 | 2.0 | 6.1 | 7.0 | 8.3 | 5.9 | 5.3 |
| EI-dia-C | 6.4 | 14.3 | 14.4 | 4.2 | 1.1 | 5.9 | 8.4 | 8.8 | 6.5 | 3.0 |
| EI-dia-M | 6.2 | 14.5 | 13.8 | 4.1 | 1.1 | 6.4 | 8.3 | 9.6 | 6.8 | 3.3 |
| E5-kin-C | 3.3 | 8.3 | 8.6 | 2.9 | 1.2 | 3.9 | 3.2 | 6.3 | 3.9 | 2.2 |
| E5-kin-M | 3.4 | 9.3 | 9.4 | 2.6 | 1.3 | 4.3 | 3.0 | 6.2 | 3.7 | 2.4 |
| E5-dia-C | 3.4 | 10.7 | 10.4 | 2.5 | 0.9 | 4.3 | 3.7 | 6.6 | 3.9 | 2.2 |
| E5-dia-M | 3.3 | 9.8 | 10.5 | 2.8 | 0.8 | 4.3 | 4.4 | 6.3 | 3.9 | 2.2 |
| E5-kin-M-ECP | 0.1 | 0.2 | 0.4 | 0.1 | 0.1 | 0.1 | 0.1 | 0.2 | 0.1 | 0.1 |
| E5-kin-M-1800s | 3.3 | 9.0 | 9.7 | 2.5 | 1.3 | 4.2 | 3.0 | 5.4 | 3.9 | 2.5 |
| E5-kin-M-Diff | 3.3 | 9.0 | 9.4 | 2.9 | 1.3 | 4.1 | 3.1 | 6.5 | 3.8 | 2.2 |
| E5lr-kin-M | 4.6 | 11.5 | 12.7 | 4.5 | 1.6 | 4.3 | 4.1 | 6.6 | 4.2 | 3.5 |
| E5lr-dia-M | 3.6 | 13.0 | 14.1 | 3.0 | 1.0 | 4.0 | 3.9 | 5.9 | 4.0 | 2.3 |





**Table E2.** Median transport time from the MB into the UTLS in days, determined by the difference between the leaving time at the MB and the arrival time above 340 K. Abbreviations are as defined in Table A1 and A2.

| Scenario | All | Res. Oc. | TWP | AS | SEA | BOB | IGP | SI | AH | Res Cont |
|---|---|---|---|---|---|---|---|---|---|---|
| EI-kin-C | 4.7 | 10.0 | 7.3 | 3.3 | 0.3 | 5.2 | 5.2 | 7.7 | 4.9 | 5.2 |
| EI-kin-M | 4.9 | 10.0 | 10.1 | 2.9 | 0.4 | 5.2 | 5.1 | 6.9 | 5.2 | 4.7 |
| EI-dia-C | 3.6 | 11.2 | 11.6 | 2.7 | 0.3 | 5.3 | 5.4 | 8.2 | 5.5 | 1.7 |
| EI-dia-M | 3.3 | 11.3 | 11.3 | 3.0 | 0.3 | 5.5 | 5.2 | 8.3 | 5.4 | 1.8 |
| E5-kin-C | 1.1 | 5.0 | 7.2 | 1.5 | 0.4 | 3.5 | 1.5 | 5.3 | 3.0 | 1.2 |
| E5-kin-M | 1.2 | 5.5 | 8.2 | 1.5 | 0.4 | 3.6 | 1.6 | 5.6 | 3.2 | 1.2 |
| E5-dia-C | 1.0 | 7.9 | 8.3 | 1.5 | 0.3 | 3.6 | 1.7 | 5.5 | 2.9 | 1.5 |
| E5-dia-M | 1.0 | 6.8 | 6.9 | 1.6 | 0.3 | 3.8 | 1.9 | 5.5 | 3.2 | 1.5 |
| E5-kin-M-ECP | 0.0 | 0.1 | 0.1 | 0.0 | 0.0 | 0.0 | 0.0 | 0.1 | 0.0 | 0.0 |
| E5-kin-M-1800s | 1.2 | 5.2 | 8.2 | 1.5 | 0.4 | 3.6 | 1.4 | 4.6 | 2.9 | 1.6 |
| E5-kin-M-Diff | 1.1 | 5.2 | 8.0 | 1.5 | 0.4 | 3.7 | 1.3 | 5.8 | 3.0 | 1.0 |
| E5lr-kin-M | 2.5 | 7.7 | 9.8 | 3.1 | 0.7 | 3.7 | 2.2 | 5.9 | 3.4 | 2.7 |
| E5lr-dia-M | 1.4 | 10.0 | 10.8 | 2.4 | 0.5 | 3.5 | 2.0 | 5.0 | 3.6 | 1.5 |



**Appendix F: Sensitivity tests for simulations employing the extreme convection parameterisation (ECP)**

The default setting of the ECP in MPTRAC relies on a CAPE value of $0 \, \mathrm{J \, kg^{-1}}$ as a threshold for triggering convection events. For our study, we found the parametrization can be improved to avoid spurious parameterized convection events over the

Persian gulf and the Red Sea. In these regions, extremely high convective inhibition, i.e. very stable low-level layers, prevent the release of the CAPE. Therefore, we implemented an additional threshold for CIN threshold, which was set to $50 \, \mathrm{J \, kg^{-1}}$ to remove unrealistic parametrized convection events over the Persian gulf. Figure F1 illustrates the impact of the parameter choices of the convection parametrization on the source identification for different threshold settings. E5-CAPE0-CIN50 is the same scenario as simulation E5-kin-M-ECP in other parts of the paper.

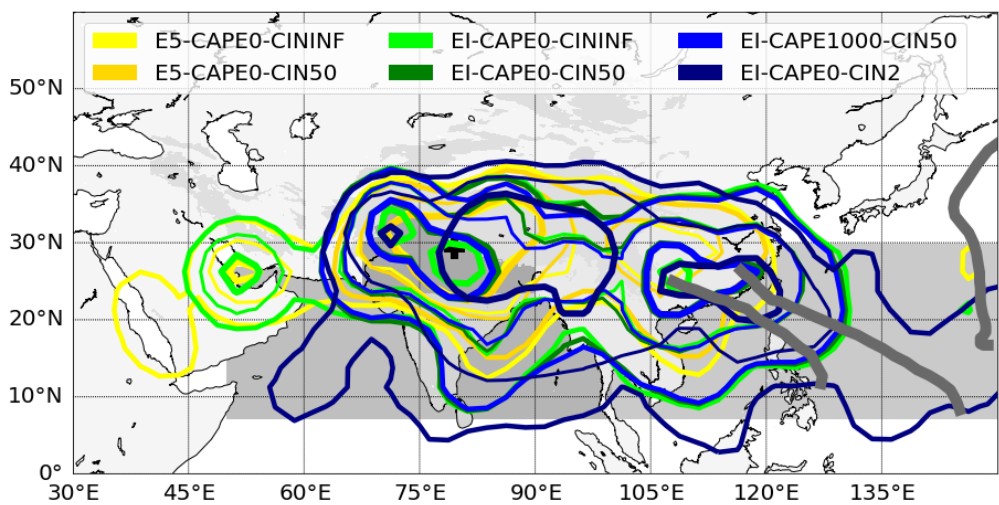

**Figure F1.** The same as in Fig. 4, but for simulations with varying settings of the ECP parameters CIN and CAPE. E5-CAPE0-CIN50 is the same as simulation E5-kin-M-ECP.



*Author contributions.* The initial idea for this study came from Bärbel Vogel and Lars Hoffmann. The formal analysis, programming and further investigations were done by Jan Clemens. Bärbel Vogel, Lars, Hoffmann, Nicole Thomas and Sabine Grießbach supervised the work and gave support with data and software. The writing was done by Jan Clemens. Coauthors contributed with review and comments.

*Competing interests.* The authors declare that no competing interests are present.

*Acknowledgements.* The authors gratefully acknowledge Simone Brunamonti, Teresa Jorge, Sreeharsha Hanumanthu, Peter Ölsner, Man-
ish Naja, Bhupendra Bahadur Singh, Kunchala Ravi Kumar, Sunil Sonbawne, Hannu Jauhiainen, Holger Vömel, Frank G. Wienhold and
Ruud Dirkson for conducting the measurements in Nainital and for providing the data. The work presented includes contributions to the
NSFC–DFG 2020 project ATALtrack (VO 1276/6-1). This research has been supported by the Helmholtz Association of German Re-
search Centres (HGF) through the projects Pilot Lab Exascale Earth System Modelling (PL-ExaESM) and Joint Lab Exascale Earth System
Modelling (JL-ExaESM). We acknowledge the Jülich Supercomputing Centre for providing computing time and storage resources on the
JUWELS supercomputer. Jan Clemens was partly funded by Helmholtz Interdisciplinary Doctoral Training in Energy and Climate Research
(HITEC). We thank the Japan Meterological Agency for providing the typhoon track data. We also thank the ECMWF for providing access
to the ERA5 and ERA-Interim reanalysis data.



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
