# Peer review of "A multi-scenario Lagrangian trajectory analysis to identify source regions of the Asian Tropopause Aerosol Layer on the Indian subcontinent in August 2016"

_EGUsphere, 2022_

## Author Comment (AC1)

**Authors Response to Referees**
**Identification of source regions of the Asian Tropopause Aerosol Layer on the Indian subcontinent in August 2016**

Jan Heinrich Clemens

May 2023

**1 Reply to Review Comment 1**

**Comment:** The paper addresses the interesting topic of the Asian tropopause aerosol layer (ATAL). The presented results provide interesting information regarding the source regions and the transport pathways of the ATAL based on balloon-borne measurements and two Lagrangian transport models driven by different reanalysis data (ERA5, ERA-interim) and model parameterisations/setups. The investigation of the ATAL is a topic of many studies during the last years. The added value of the present paper is the thorough intercomparison between the results obtained with different simulation scenarios (with different Langrangian model employed, the use of different reanalysis data -ERA5 vs ERA-interim-, vertical coordinate -kinematic vs diabatic approach- and convective parametrisation).
In my opinion, since the publication focuses mainly on the comparison between the different simulation scenarios, this should be reflected in the title. Overall the manuscript is well written and I recommend its publication in ACP after providing some clarifications and addressing some minor issues.

**Answer:** Dear Reviewer No. 2, many thanks for your positive review and your remarks. We agree, that the title of the paper should reflect the strength of our study more. Therefore, the title now is: *"A multi-scenario Lagrangian trajectory analysis to identify source regions of the Asian Tropopause Aerosol Layer on the Indian subcontinent in August 2016"*

**Comment:** I think that it is better to use the term "model boundary layer" instead of the term "model boundary". (at least when it is mentioned for the first time).

**Answer:** We now use the term "model boundary layer" instead of "model boundary".

**Comment:** The Asian highlands is not a continuous region. It would be interesting to mention the percentage of air masses with origin only from the Thibetan Plateau.

**Answer:** We added extra information about the Tibetan plateau (TP). In the definition of the regions we added the size of the TP (plus immediate adjacent regions) relative to the other Highlands (around 60%). We added the information that 95% to 98% (or with a very narrow definition of the TP: 72% to 82%) of the air parcels that originate from the Highlands are from the TP depending on the scenario (see page 8, line 206-210). Finally, we checked wether the narrow definition of the TP changes results with regard to the correlation. Since it didn't we added a remark for this in the paper (see page 22, line 439-440).

**Comment:** l45: "up to 360K". It would be useful to clarify here (for readers less familiar with the subject) that you are referring to potential temperature.

**Answer:** We have added this information: *"...(up to a potential temperature height of about 360K)"* (see page 2, line 45-46).

**Comment:** l l120-122 "The hybrid coordinate $\zeta$ ... above around 300 hPa and 380 K, respectively. "Please rephrase to "The hybrid coordinate $\zeta$ is near the surface an orography-following sigma coordinate and transforms continuously into potential temperature at higher altitudes above around 300 hPa (or 380 K).

**Answer:** We have rephrased this sentenced accordingly.

**Comment:** l229 "transport pathways from two regions with two very different land-cover properties converge, the Tibetan Plateau or the Indo-Gangetic plain." please correct to "Tibetan Plateau and the Indo-Gangetic plain"

**Answer:** We have corrected this as well.

**Comment:** l240 and Fig. 1d: please correct the typo "Arabian See" -¿ "Arabian Sea"

**Answer:** Fixed.

**Comment:** l245 "from the Bay of Bengal more air parcels can enter the ASMA directly from the maritime boundary than from the Arabian Sea". Can you elaborate on this and provide an explanation? Also why the mean transport time from the MB into the UTLS is higher compared to the AS?

**Answer:** With regard to the second question: Thanks to your remark, we spotted a formal mistake in the table. With the correct order of labels, it can be seen now that transport from the AS into the UTLS is slower than from the BoB, in accordance with the longer transport pathway. We as well have corrected some numbers in the text for this reasons.

To the first remark: We added: *"This is likely related to the overall Asian monsoon circulation, in which air masses are transported in the troposphere, from the AS across India, while the Bay of Bengal is a source of deep convection. Moreover, those air masses that are convectively uplifted into the UTLS over the AS, are often located at the outer edge of the anticyclone and westward from Nainital. Hence, this transport pathway to Nainital is much less probable."* to give further context of the difference between the BoB and the AS, in relation to the circulation and the trajectory analysis (see page 10, line 264-268).

**Comment:** ll253-254: "Wide agreement can be found with ERA5 even when models, integration step-sizes, and vertical velocities are varied." and l268: "Model scenarios driven with ERA5 show very similar results." The scenario with extreme convection parameterisation (E5-kin-M-ECP) is also driven by ERA5 reanalysis data. It's better to mention this here (e.g. Model scenarios driven with ERA5 show very similar results except when employing the extreme convection parameterisation)

**Answer:** We rephrased to clarify that we do not refer to the scenarios with parameterisation: *Wide agreement can be found with ERA5 even when models, integration step-sizes, and vertical velocities are varied (except when the extreme convection parameterisation is employed).* (see page 12, line 275-276).

**Comment:** According to Figs 3b and 4c under the scenario with ECP the contributions from SE Asia increase. Please elaborate more on this in the text.

**Answer:** *As a consequence of the frequent occurrence of CAPE above South Asia, the scenario with ECP simulates more and deeper convective updrafts in this region, than the scenarios without ECP. Hence, transport from the MBL that would be missed without the ECP, increases the contributions from South Asia.*, were added (see page 12, line 301-302).

**Comment:** ll300-301: " This leads to... Indo-Gangetic plain." Please elaborate more on this.

**Answer:** We added: *When air parcels passing the region around the Himalayas during the backward trajectory calculations, the position where they are transported back into the MBL is sensitive to the representation of the convection. With the ECP, convection is enhanced over the IGP. Hence, far more air masses are attributed to the IGP with the ECP, while without the ECP the most air parcels originate from the TP. Further improvement in reanalysis and parameterisation are needed to remove this uncertainty.* (see page 13, line 326-328).

**Comment:** ll313-314: Please rephrase (begin the sentence with "For ERA-Interim...")

**Answer:** We have rephrased this sentence.

**Comment:** l322: " However, our approach of the ECP has to be considered as an upper limit for convective transport." Can you elaborate more on this? In the Appendix F You mention that "the parametrization can be improved to avoid spurious parameterized convection events over the Persian gulf and the Red Sea". It was not possible to further imporove the parametrization? I mention this because I am concerned about the large differences between the scenario with ECP and the other scenarios.

**Answer:**
We agree that the statement of an upper limit for convection was to strong, as the true strength of convection is not known. Therefore we reformulated more carefully: *However, our approach of the ECP has to be considered as the upper limit for the convective transport that can be simulated within the given model framework.* and provided in the data and methods section more context for the ECP:

*The ECP was introduced by Gerbig et al. [2003] to estimate the upper limit of convective cloud transport and later was as well implemented in HYSPLIT [Loughner et al., 2021]. The ECP vertically mixes the air parcels within a convective column by a randomized density-weighted distribution between the surface and the equilibrium level (EL). Hence, it is assumed, that the vertical column is perfectly mixed after a convective event, so that further mixing could not change the distribution anymore. [...] The parameterisation scheme was configured for the largest possible enhancement of convective transport, additionally to the already resolved convection in the reanalysis. The CAPE threshold for the triggering of the parameterisation was set to 0 $\text{Jkg}^{-1}$ for that purpose. The combination of perfectly vertical mixing and the most sensitive configuration of the trigger parameter, allows for estimating the upper limit for the convective transport that can be simulated within the given model framework.*

We want to emphasize, that the focus of our study is to expose and quantify exactly this concerning differences within our given Lagrangian transport framework, but as well, on the other side, to point out the most robust findings despite the differences. In particular after the intercomparison we did, we strongly agree that the convection parameterisation needs further development and evaluation with observational data. Moreover, we think that the truth lies somewhat between the ECP scenario and the ERA5 scenarios. With current and future developments we want to address exactly this remaining uncertainties that our analysis shows [see also Hoffmann et al., 2023].

**Comment:** Appendix F: Please provide a table with a short description of the scenarios shown in Fig F1. (similar to Table 2)

**Answer:** We added the Table F1 with a short description of the scenarios.

**Comment:** parametrization vs parameterisation: Please choose one spelling for consistency.

**Answer:** Done.

**2 Reply to Review Comment 2**

**Comment:** This paper's goal is to trace the source regions and pathways of air that forms the ATAL over Nainital, India in August 2016 using two different Lagrangian transport models (CLaMS MPTRAC) utilizing two different meteorology sources (ERA-Interim ERA5), and two different vertical coordinates (kinematic and diabatic) to create an ensemble of results to measure the robustness source locations. This paper adds significant value in that it compares all of the results of these models, meteorology sources, and vertical coordinates to identify the most likely source regions and pathways of air that forms the ATAL.
I fully agree with the other reviewers comment that since this study focuses so heavily on the comparison of different simulation scenarios, the modeling and transport aspects of this work should be reflected in the paper title in some way.
I think this paper is well written and thoughtfully laid out and recommend publication in ACP after a few very minor clarifications and revisions.

**Answer:** Dear Reviewer No. 1, many thanks as well for your encouraging review and your remarks. We have changed the title paper to *"A multi-scenario Lagrangian trajectory analysis to identify source regions of the Asian Tropopause Aerosol Layer on the Indian subcontinent in August 2016"*.

**Comment:** Line 61-62, it may be better to say that 'These calculations rely on reanalysis data and their ability to adequately resolve convection and diabatic vertical ascent in the ASMA.' since no model 'correctly' resolves convection.

**Answer:** As we agree that no reanalysis correctly can resolve the convection, we changed to the more careful word "adequat".

**Comment:** Line 84, It doesn't appear that the authors are dealing with any explititly modeled convection in this paper. The convection in all of the models described in this paper use parameterized convection.

**Answer:** We agree that all the used reanalysis do not explicitly model convection and that parameterisation is applied as well in the reanalysis. However, here we refer to parameterisation within the Lagrangian transport models. We added a paragraph into the data and methods section to clarify the difference between the two parameterisation and the motivation: *Atmospheric models that are used to create reanalysis data need to apply convective parameterisation to mitigate limitation in the resolution of convection. However, since the reanalysis product does only contain the averaged velocities, Lagrangian transport models have to employ a convective parameterisation as well. Therefore, the extreme convection parameterisation (ECP) has been implemented recently into the MPTRAC to represent the effects of unresolved convection in the reanalysis data [Hoffmann et al., 2023].* (see page 5, line 133-135)

**Comment:** Line 133-139, Could reference Appendix F in this paragraph.

**Answer:** We have restructured the paragraphs to address some other comments, and added the references at the end (see page 5, line 152-154)

**Comment:** Line 210, can also get a third - albeit less frequent - ASMA mode over the western Pacific Ocean near Japan (from Honomichl and Pan, 2020).

**Answer:** Thanks for this reminder. We added this information (see page 9, line 230-231)

**Comment:** Line 317, maybe a better way to say is that the parameterization of convection performs better in ERA5 than it does in ERA-i

**Answer:** We agree that the parameterisation likely plays an important role for the improvements, but, we as well think that the increased resolution in ERA5 improves the simulations. Since we cannot say with enough certainty, we reformulated it now more general to: *"This is likely due to a better representation of convection in ERA5 in comparison to ERA-Interim."*.

**Comment:** Line 319-320, Is the ERA5 convective parameterization missing the whole onset of convection or is the convection present but just as deep compared to the ECP? The additional information could be a good Segway to the comment that ECP is considered an upper limit for convective transport.

**Answer:** Please see the answer to reviewer no. 2, on page 9

**Comment:** Line 368, several 10% is a little confusing. I'm not entirely sure what you mean by it.

**Answer:** We now write *"From day to day, the relative normalized deviation changes in absolute terms with a magnitude of only a couple 10%. This is in contrast to the total change of the relative normalized deviation during the period, which is roughly around 90%."* to clarify that the percentage refers to the unit of the relative normalized deviation. (see page 9, line 370-375)

**Comment:** Line 451, parameterizes convection better than ERA-i

**Answer:** Fixed.

**Comment:** Line 451, "it might still underestimate"

**Answer:** Fixed.

**References**

C. Gerbig, J. C. Lin, S. C. Wofsy, B. C. Daube, A. E. Andrews, B. B. Stephens, P. S. Bakwin, and C. A. Grainger. Toward constraining regional-scale fluxes of

CO2 with atmospheric observations over a continent: 2. analysis of COBRA data using a receptor-oriented framework. *Journal of Geophysical Research*, 108(D24), 2003. doi: 10.1029/2003JD003770.

Christopher P. Loughner, Benjamin Fasoli, Ariel F. Stein, and John C. Lin. Incorporating Features from the Stochastic Time-Inverted Lagrangian Transport (STILT) Model into the Hybrid Single-Particle Lagrangian Integrated Trajectory (HYSPLIT) Model: A Unified Dispersion Model for Time-Forward and Time-Reversed Applications. *Journal of Applied Meteorology and Climatology*, 60(6):799–810, June 2021. ISSN 1558-8424, 1558-8432. doi: 10.1175/JAMC-D-20-0158.1. URL https://journals.ametsoc.org/view/journals/apme/60/6/JAMC-D-20-0158.1.xml.

L. Hoffmann, P. Konopka, J. Clemens, and B. Vogel. Lagrangian transport simulations using the extreme convection parametrization: an assessment for the ecmwf reanalyses. *EGUsphere*, 2023:1–29, 2023. doi: 10.5194/egusphere-2023-72. URL https://egusphere.copernicus.org/preprints/2023/egusphere-2023-72/.

---

## Author Response (AR2)

**Authors Response to Referees**

**A multi-scenario Lagrangian trajectory analysis to identify source regions of the Asian Tropopause Aerosol Layer on the Indian subcontinent in August 2016**

Jan Heinrich Clemens

Oktober 2023

**Comment:** Abstract: Your abstract has current $\tilde{4}20$ words. According to ACP's new author guidelines, we recommend abstracts of 250 words. https://www.atmospheric-chemistry-and-physics.net/policies/guidelines_for_authors.html While I understand that you submitted your paper way before the implementation of these guidelines, I encourage you to shorten the abstract to focus on the main points and concisely describe topic/status of knowledge/gap/objectives/approach/results/importance.

**Answer:** Many thanks for handling the editing process and giving comments as Editor. To shorten the abstract we have removed some information of minor importance and used shorter formulations.

**Comment:** l. 36: 'originally' should be replaced by 'initially'

**Answer:** Done.

**Comment:** l. 52: 'was shown' should be 'were shown'

**Answer:** Done.

**Comment:** l. 129: Please use consistently either 'zeta' or the Greek symbol.

**Answer:** We have replaced zeta coordinates by the Greek symbol where ever possible.

**Comment:** l. 133: 'mitigate limitation in the resolution on convection' sounds awkward. Do you mean 'since they cannot resolve the small-scale convection processes'?

**Answer:** We rephrased the sentence to clarify its meaning: *Atmospheric models that are used to create reanalysis data need to apply convective parameterisation, because they are not able to resolve the small-scale convection processes.*

**Comment:** l. 134/5: replace 'does only contain' by 'only contains'

**Answer:** Done.

**Comment:** l. 152: Text seems redundant here. Can 'It will be explained in more detailed later' be removed? If it is explained in another section, in addition to appendix F1, please add the section number here.

**Answer:** We added the reference to the section 3.2., which is relevant to understand the selection of CIN.

**Comment:** l. 180 'each' can be omitted

**Answer:** Yes and done.

**Comment:** l. 245/6: Is it 'plain' or 'plains'?

**Answer:** We changed it to plain.

**Comment:** l. 259: Arabian Sea

**Answer:** Done

**Comment:** l. 329: delete 'the' (..the most..)

**Answer:** Done

**Comment:** l. 337: What do you mean by 'maritime processes'?

**Answer:** Maritime processes should refer to convection over the oceans and seas that transport the air upward into the UTLS. To clarify this we added: *Most of the relevant maritime convection (e.g. typhoons) that transport air masses out of the MBL into the upper atmosphere take place more than two weeks before the measurements.*

**Comment:** l. 382: (1) please write the equation in a separate line and give it a number, according to the mathematical notation and terminology guidelines https://www.atmospheric-chemistry-and-physics.net/submission.html (2) Make sure that the bar above the denominator can be clearly distinguished from the fraction stroke. Depending on screen resolution, currently it looks like C(t)/C(t) separated by a thick line. You may want to consider using / instead of the horizontal separation.

**Answer:** We have followed your suggestions and added a number for the equation.

**Comment:** l. 398: What do you mean by 'a couple 10%'? 'a couple' is synonymous with two, i.e. 20%? Or could you replace it by 'approximately 10%' or 'several tens of percent'?

**Answer:** We refer to changes between 10% and 100%. To clarify this we rephrased: *From day to day, the relative normalized deviation changes in absolute terms with the order of magnitude of 10% (i.e. 10-100%), while the total change of the relative normalized deviation during the period is roughly 90%.*

**1   Reply to Reviewers Comment**

**Comment:** My previous remarks were properly addressed by the authors. I have only a few minor comments, listed below.

**Answer:**   Thank you for the review of our changes and your further comments.

**Comment:** l 122: I think that the world respectively can be omitted.

**Answer:** Yes, we removed it.

**Comment:** ll 310-311: "As a consequence of the frequent occurrence of CAPE" Do you mean frequent occurrence of high CAPE values?

**Answer:** We rephrased to *"As a consequence of the persistent occurrence of CAPE above South Asia in summer, the scenario with ECP simulates more and deeper convective updrafts in this region, than the scenarios without ECP."* However it is indeed the occurence of CAPE that triggers the parameterisation. The height of CAPE does not affect the parameterisation in the chosen set-up, because the CAPE threshold is 0. See also Hoffmann et al. [2023] Fig. 10 for further details on the occurrence of CAPE.

**Comment:** please correct the following typos: l 267: Arabian see → Arabian Sea l 153 and l 330: Persian Golf → Persian Gulf

**Answer:** Done.

**2 Further changes**

**Comment:** We found an error in the plotting routine for Figure 5a.

**Answer:** We corrected the plot and removed: *"except for the scenario with ECP."* (line 339), while no other part of the manuscript and analysis has been affected.

**References**

L. Hoffmann, P. Konopka, J. Clemens, and B. Vogel. Lagrangian transport simulations using the extreme convection parametrization: an assessment for the ecmwf reanalyses. *EGUsphere*, 2023:1–29, 2023. doi: 10.5194/egusphere-2023-72. URL https://egusphere.copernicus.org/preprints/2023/egusphere-2023-72/.